# Pervasive glacier retreats across Svalbard from 1985 to 2023

Tian Li [1,2] ✉, Stefan Hofer[2], Geir Moholdt [3], Adam Igneczi[2], Konrad Heidler [1], Xiao Xiang Zhu [1,4] & Jonathan Bamber [1,2]

A major uncertainty in predicting the behaviour of marine-terminating glaciers is ice dynamics driven by non-linear calving front retreat, which is poorly understood and modelled. Using 124919 calving front positions for 149 marine-terminating glaciers in Svalbard from 1985 to 2023, generated with deep learning, we identify pervasive calving front retreats for non-surging glaciers over the past 38 years. We observe widespread seasonal cycles in calving front position for over half of the glaciers. At the seasonal timescale, peak retreat rates exhibit a several-month phase lag, with changes on the west coast occurring before those on the east coast, coincident with regional ocean warming. This spatial variability in seasonal patterns is linked to different timings of warm ocean water inflow from the West Spitsbergen Current, demonstrating the dominant role of ice-ocean interaction in seasonal front changes. The interannual variability of calving front retreat shows a strong sensitivity to both atmospheric and oceanic warming, with immediate responses to large air and ocean temperature anomalies in 2016 and 2019, likely driven by atmospheric blocking that can influence extreme temperature variability. With more frequent blocking occurring and continued regional warming, future calving front retreats will likely intensify, leading to more significant glacier mass loss.

Svalbard is located in the most rapidly warming region of the Arctic and experiences frequent climate extremes[1–4]. It is adjacent to the main pathway of the warm Atlantic Water towards the Arctic Ocean carried by the West Spitsbergen Current[5] and in the path of atmospheric rivers in the North Atlantic sector[3]. ~57% of the Svalbard area is covered by glaciers[6,7] which are at relatively low elevation with flat interior accumulation areas, making mass loss of these glaciers strongly sensitive to climate warming[8–12]. Glaciers in Svalbard have been losing mass at accelerated rates during the past several decades[13–16], their mass loss is expected to double by 2100 but with significant uncertainties in the predicted rates[11], partly due to a lack of consideration of retreats of marine-terminating glaciers[7,17] which drain ~69% of the glacierized area[18]. Therefore, it is crucial to examine how glacier retreat

at the ice-ocean boundary responds to different climate drivers across Svalbard.

Significant calving front retreats of marine-terminating glaciers have been identified for ice masses across the Arctic, which feature diverse atmospheric, oceanic and glaciological settings[19–29]. However, detailed analyses of calving front change mechanisms have primarily focused on the Greenland Ice Sheet[19–22] and its periphery glaciers[23–25]. So far, only two studies have quantified the frontal ablation (including solid ice calving and frontal melting at the glacier calving front) of all Svalbard glaciers, either over a short period from 2000 to 2006[30] or with a low temporal resolution on a decadal scale[31]. There is a very limited understanding of the long-term and seasonal calving front changes of marine-terminating glaciers across Svalbard since the

[1]Chair of Data Science in Earth Observation, Department of Aerospace and Geodesy, Technical University of Munich, Munich, Germany. [2]Bristol Glaciology Centre, School of Geographical Sciences, University of Bristol, Bristol, UK. [3]Norwegian Polar Institute, Tromsø, Norway. [4]Munich Center for Machine Learning, Technical University of Munich, Munich, Germany. ✉e-mail: tian.li@bristol.ac.uk

satellite era, due to a lack of a high spatiotemporal-resolution calving front dataset with a large spatial scale[32]. In addition, detailed air-ice-ocean interactions have only been documented for a few marine-terminating glaciers in this region[33–36]. As a result, the climate drivers and the governing processes for calving front dynamics remain largely unknown. Here we use a calving front data product over large spatial scales and at a high temporal resolution for 149 marine-terminating glaciers across Svalbard between 1985 and 2023, mapped from a deep learning model, to investigate calving front change dynamics and its relation to air-ice-ocean interactions at different timescales over the past four decades. We first assess seasonal front retreat rates to climate changes across different parts of the archipelago. We then link the interannual calving front variability to climate extremes in recent years and investigate the dynamic mass loss likely associated with calving front retreats of marine-terminating glaciers.

## Results and discussion
### Widespread seasonal calving front changes

With the dense calving front observations generated with deep learning, we identify widespread seasonal cycles in calving front changes for marine-terminating glaciers across entire Svalbard over the period of 2014–2023 (Fig. 1a), a phenomenon that was only documented for a handful of glaciers in this region[33–39]. 86 out of the 138 (62%) non-surging marine-terminating glaciers have seasonal cycles (annual periodicity autocorrelation ≥ 0.08, see Methods). These

glaciers are mainly located on the western side of Svalbard, including sectors Northwest and South Spitsbergen that are in close contact with warm ocean currents (Fig. 1a), as well as the northeastern part of Svalbard, such as Austfonna (Fig. 1a) that is featured by complex oceanography settings with a mix of warm Atlantic Water and cold water from the Arctic Ocean and the Barents Sea[40,41]. The mean seasonal range along the glacier centerline for these 86 glaciers is 102 ± 6 m. At the regional scale, South Spitsbergen has the highest mean seasonal range of 141 ± 3 m, followed by Northwest and Vestfonna (Figs. 1a and 2a). In addition, glaciers Olsokbreen (RGI60-07.00265, Figure S3) and Paierlbreen (RGI60-07.00244, Figure S4) with the highest seasonal ranges of 411 m and 385 m respectively are located in South Spitsbergen. We consider calving front advance as when ice discharge exceeds frontal ablation and retreat as when frontal ablation exceeds ice discharge. We are not able to quantify ice discharge, and therefore frontal ablation, due to a lack of ice velocity dataset of sufficient spatial coverage and temporal resolution for the entire Svalbard over a long period that is required for this study. However, calving front retreat rate has proved to be a good indicator for the variability of frontal ablation in Svalbard[35,36]. In these studies where ice discharge has been taken into account, it is shown that frontal ablation in winter almost shuts down, allowing most of the ice discharge to go into the seasonal readvance of the calving fronts.

Across Svalbard, glaciers generally start retreating from May to July, reach the highest retreat rates in August and September, and

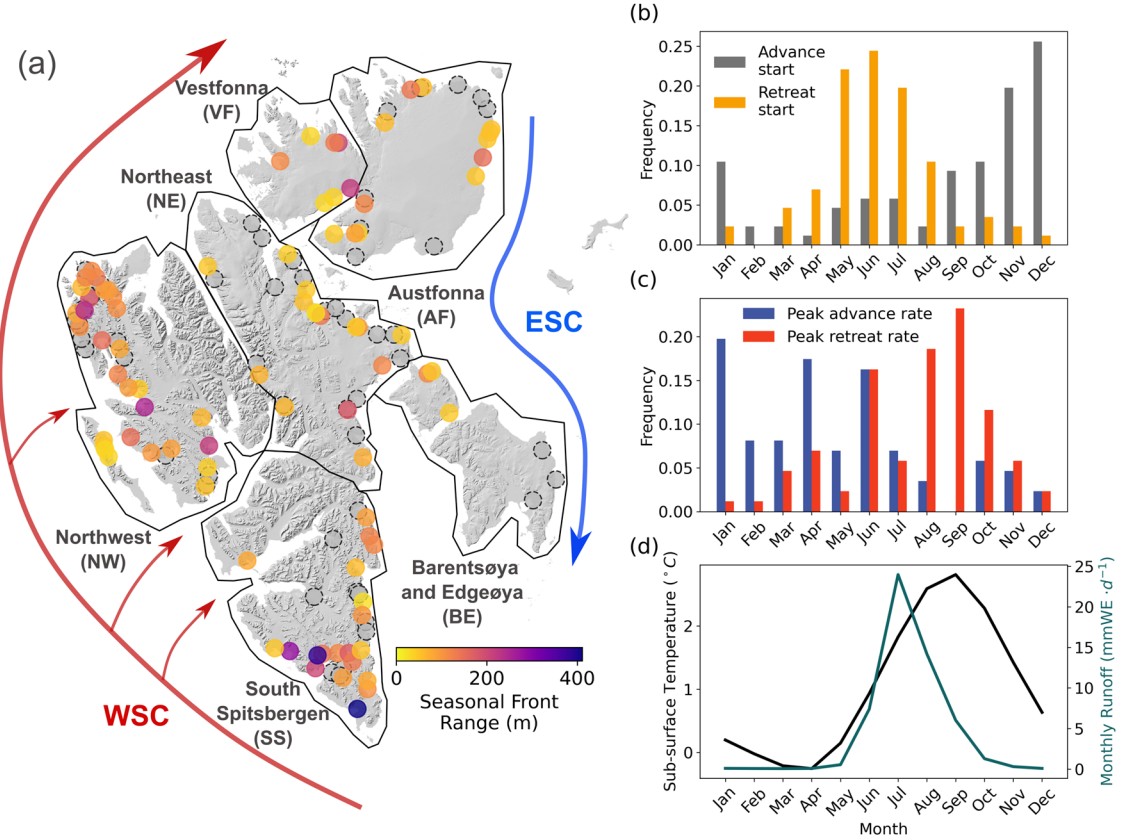

**Fig. 1 | Seasonal calving front changes. a** Coloured circles denote the seasonal calving front range along the glacier centreline for 86 non-surging marine-terminating glaciers with seasonal cycles (autocorrelation ≥ 0.08). Dashed circles denote the non-surging glaciers that do not display a seasonal cycle. The red and blue arrows show the routes of the warm West Spitsbergen Current (WSC) and the cold East Spitsbergen Current (ESC), respectively. Solid black polygons show the major sectors of Svalbard defined in this study, including Northeast Spitsbergen (NE), Northwest Spitsbergen (NW), South Spitsbergen (SS), Vestfonna (VF), Austfonna (AF), and Barentsøya and Edgeøya (BE). The background hillshade map is generated

from the 50 m resolution Svalbard digital elevation model (DEM) (https://data.npolar.no/dataset/dce53a47-c726-4845-85c3-a65b46fe2fea, last access: 18 April 2023). **b** Monthly frequency of glacier advance onset (grey) and retreat onset (yellow). **c** Monthly frequency of the peak calving front advance rate (blue) and retreat rate (red). **d** Monthly mean surface runoff from the MAR model (solid green line) and monthly mean 20–100 m depth-averaged sub-surface ocean temperature from the ORAS5 ocean reanalysis data (solid black line). Source data are provided as a Source Data file.

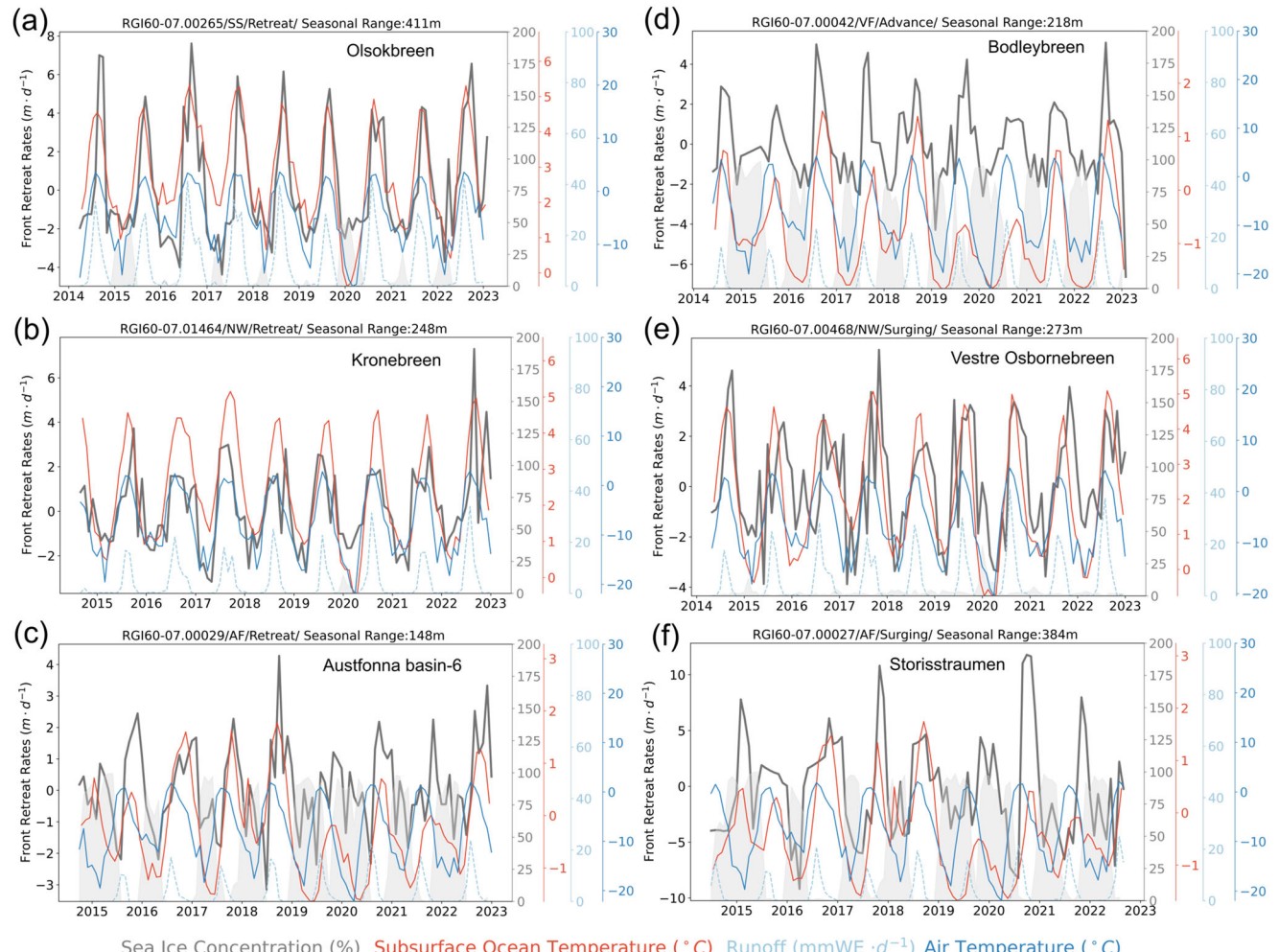

Fig. 2 | Glacier front time series. Highlight glaciers' time series of detrended retreat rates (solid black line), sea ice concentration (shaded grey area), subsurface ocean temperature (solid red line), surface runoff (dashed blue line), and air temperature (solid blue line). a–c Retreating glaciers located in South Spitsbergen (SS), Northwest Spitsbergen (NW) and Austfonna (AF), d advancing glacier located in Vestfonna (VF), e, f surging glaciers located in NW and AF. The title of each panel includes information of the RGI60 glacier ID, the sector of the glacier's location, the type of calving front change, and the seasonal range.

then start to re-advance in November and December (Fig. 1b-c). Examples of the seasonal calving front change time series of several non-surging glaciers are shown in Fig. 2a–d. The calving front locations display regular seasonal cycles with the peak retreat rates occurring around the same time of the year. The onset of calving front retreat coincides with the increased surface runoff and rising subsurface ocean temperature, while the peak retreat rates occur during the months of highest ocean temperatures (Fig. 1d). This is in agreement with a recent study focusing on outlet glaciers of Austfonna[36] which discovered synchronised seasonal cycles in calving front changes in phase with nearby ocean temperature measurements in 2019–2022. We identified eight glaciers across Svalbard having seasonal cycles during a glacier surge (Table S2 and Figure S9a). The seasonal calving front changes of surging glaciers follow similar patterns to non-surging glaciers (Figure S9b), although their average seasonal range is significantly higher, reaching $192 \pm 54$ m. Figure 2e, f show the seasonal calving front time series of two surging glaciers Vestre Osbornebreen (RGI60-07.00468) and Storisstraumen (RGI60-07.00027), respectively, where calving fronts usually retreat from July until December with a regular cycle. The similarity in seasonal cycles between surging and non-surging glaciers provides further evidence that dynamic regimes have minimal influence on the calving front cyclicity[35].

## Ocean temperature variability drives divergent seasonal cycles

The coincidence between peak seasonal retreat rates and peak subsurface ocean temperatures indicates the impact of ocean heat content on seasonal calving front retreat cycles. Correlation analysis between seasonal calving front retreat rates with four different time-evolving environmental variables including sea ice concentration, surface runoff, 3 m air temperature, and 20–100 m depth-averaged subsurface ocean temperature, shows that ocean temperature is the strongest predictor for seasonal calving front rates with an $R^2$ value of 0.97 (Figure S7). Similar relationships are also found for surging glaciers (Figure S10). Northwest and South Spitsbergen have the highest correlations (Table S1), where the 20–100 m depth-averaged subsurface ocean temperature is a mixture of surface water and Atlantic Intermediate Water[5]. The spatial distributions of the fjord points used to extract ocean temperatures from ORAS5 ocean reanalysis data are shown in Figure S6. Although some fjord points are far from glacier terminus and sample ocean temperatures outside fjords due to limited spatial resolution and coverage of ocean reanalysis data near the coast, this does not significantly affect the relationship between ocean warming and seasonal calving front retreats. The fjord points in the open ocean can still be representative of the water structures within the fjord, as they are predominantly located in regions of deep bathymetry (Figure S6), indicating that these fjord points lie along the pathway where warm Atlantic water can enter the fjords. Furthermore,

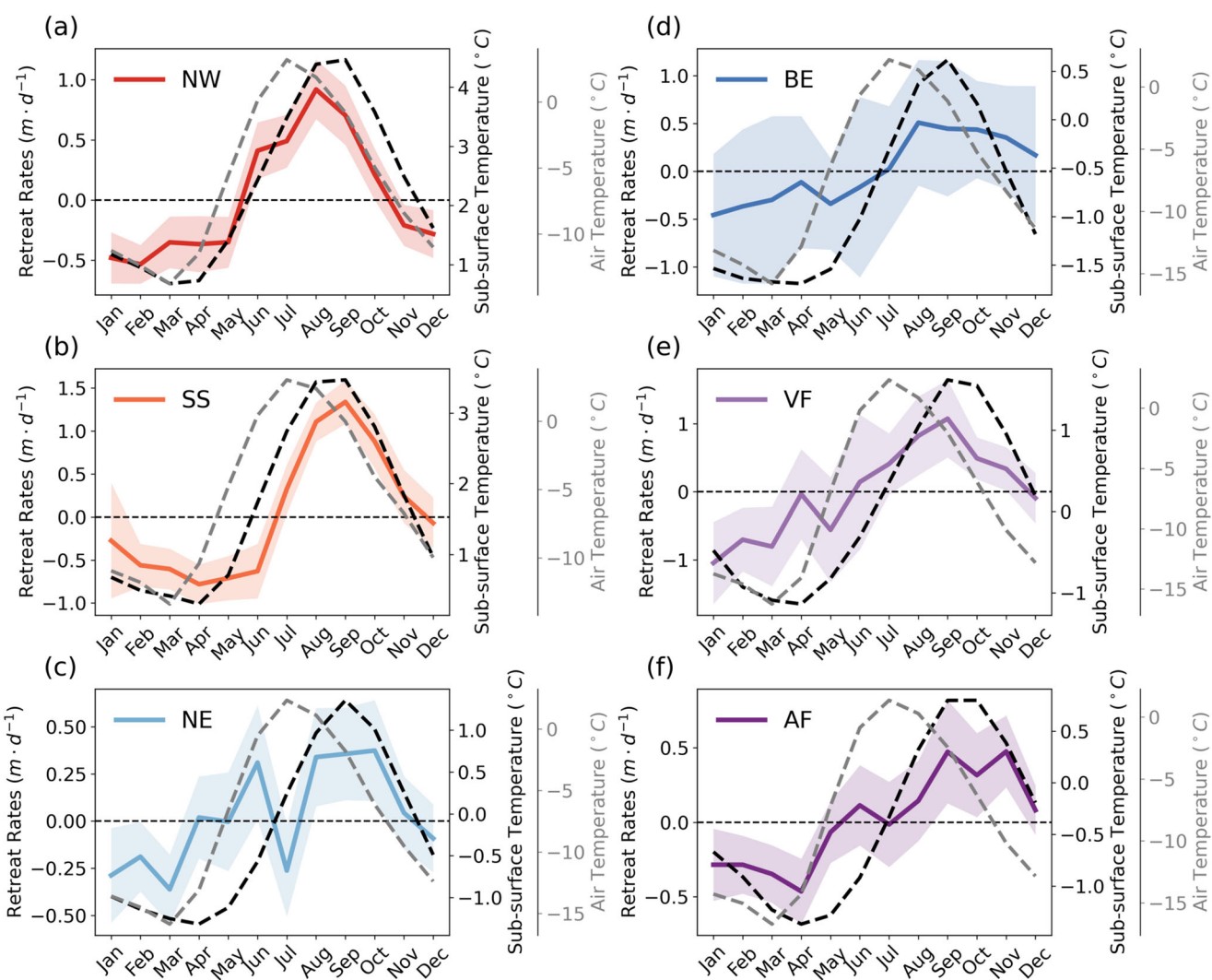

**Fig. 3 | Regional variations in seasonal retreat rates.** The monthly mean detrended retreat rates for non-surging glaciers with seasonal cycles in each region overlaid with monthly mean subsurface ocean temperature (black dashed line) and monthly mean air temperature (grey dashed line): (**a**) Northwest Spitsbergen (NW), **b** South Spitsbergen (SS), **c** Northeast Spitsbergen (NE), **d** Barentsøya and Edgeøya (BE), **e** Vestfonna (VF), **f** Austfonna (AF).

in-situ measurements in west Spitsbergen show that ocean temperatures in water depths less than 100 m are often similar both inside and outside the fjords[5,42].

From western to eastern Svalbard, there is a several-month delay in the timings of peak retreat rates and peak subsurface ocean temperatures (Fig. 3). On the west coast, the peak retreat rates typically occur in August and September, such as in the regions of Northwest (Figs. 2b and 3a) and South Spitsbergen (Figs. 2a and 3b). Meanwhile, along the east coast, peak retreat rates normally occur from September to November, such as Vestfonna (Figs. 2d and 3e) and Austfonna (Figs. 2c and 3f). This regional variation in peak retreat rates closely aligns with the seasonal subsurface ocean temperature patterns across different regions, suggesting that ocean temperature exerts an immediate impact on calving front retreat, as oceanic forcing can manifest the submarine melting of glaciers and increase glacier retreat rates by combined undercutting and calving[36,43–46]. In contrast, the peak air temperature and surface runoff consistently occur in July, irrespective of spatial location. The seasonal ocean warming is likely caused by the seasonal inflow of warm Atlantic ocean water, which normally enters fjords in an annual cycle during the shift from winter to summer and can also be present during winter due to the increased winter cyclones passing through the Fram Strait[5]. Peak retreat rates and peak ocean temperature are delayed by one-two months in

Vestfonna and Austfonna compared to western Svalbard, matching the delayed ocean heat transport of the warm West Spitsbergen Current (WSC) along the northeastern coast (Fig. 1a). In addition, the earlier onset of seasonal ocean warming and higher subsurface ocean temperatures in Northwest compared to South Spitsbergen are consistent with in-situ ocean measurements[5], which indicate that the intensity of ocean water exchange in the Northwest fjord is nearly twice that of the South Spitsbergen fjord, highlighting the significant role of Atlantic water transport in driving this difference.

Sea ice concentration ($R^2 = 0.83$) is the second strongest predictor for seasonal front retreat rates. Landfast sea ice (hereafter referred to as fast ice) can exert a buttressing effect on glacier calving fronts, stabilising the ice and reducing calving rates[47]. However, in Svalbard, there is a large spatial variability in seasonal sea ice concentration (Figure S8), and glaciers in Northwest and South Spitsbergen are largely sea ice free in winter over the 2014–2023 period (Fig. 2a, b, e). Eastern Svalbard including Austfonna, Vestfonna, Barentsøya and Edgeøya, feature higher sea ice concentration. The average sea ice concentrations in early Spring around Austfonna and Vestfonna are around 70%, while it can exceed 80% around Barentsøya and Edgeøya (Figure S8), which is the region that has the highest correlation with sea ice (Table S1). This spatial pattern matches the fast ice distribution in the past two decades which is primarily concentrated along the

coast of eastern Svalbard[48]. We also found positive correlations for seasonal retreat rates with air temperature ($R^2 = 0.63$) and surface runoff ($R^2 = 0.3$) (Figure S7), indicating that they are likely contributing factors to seasonal variations in the glacier front. Surface meltwater caused by atmospheric warming can increase subglacial discharge which can lead to increased frontal melting, as it can create buoyant plumes that enhance the turbulent transfer of oceanic heat to glacier calving fronts[21,29,45,46,49], while accumulation of meltwater on glacier surface contributes to crevasse hydrofracturing[50–52]. The surface melt season in Svalbard usually lasts about six months between May and October (Fig. 1d) and the onset of surface runoff in May coincides with the most frequent starting month for frontal retreat (Fig. 1b–d).

## 38 years of pervasive glacier retreats

After removing seasonal variability from the 1985–2023 calving front change time series (Figure S1), we found pervasive glacier retreats over the past 38 years across all six sectors (Fig. 4a), and for 91% of the 138 non-surging marine-terminating glaciers considered in this study (Fig. 4b). All non-surging glaciers lost a total area of $810 \pm 26~km^2$ at the calving front over the past four decades, equivalent to an annual areal loss rate of $23.78~km^2 \cdot yr^{-1}$ (Fig. 4c). The area change is calculated by multiplying the calving front length change by the glacier width, defined as the mean length of all terminus traces (see Methods). A comparison of the area loss of these glaciers with another calving front change dataset[53] in two different periods 2000–2010 and 2010–2020 is shown in Figure S11 (negative values denote area loss). The two datasets show good agreement, particularly during the period from

2010 to 2020 ($R^2 = 0.83$). The total area loss estimated by Kochtitzky and Copland (2022)[53] in 2000–2020 was $456~km^2$, while the net area loss is $634~km^2$ in our study. This difference is mainly caused by a discrepancy in the temporal resolution of calving front mapping in 2000–2010 for five large glaciers including Braasvellbreen, Polakkbreen, Ordonnansbreen, Stonebreen, and Hochstetterbreen (Figure S12). A slight difference in their calving front location can lead to significant area changes due to the glacier's large width. Especially, the calving front mapping frequency in Kochtitzky and Copland (2022)[53] is only 10 years which can be influenced by seasonal calving cycles compared to our long-term trend. After removing these five glaciers in the comparison, the area change difference in 2000–2020 is $49~km^2$.

The range of seasonal calving front variability correlates with the decadal retreat rates with a correlation coefficient $R$ value of 0.6 ($P < 0.05$, Figure S13a), suggesting that glaciers experience a larger amplitude in the seasonal cycle also experience a greater amplitude of secular trend. For example, glaciers Olsokbreen (Figure S3) and Paierlbreen (Figure S4) have high seasonal ranges of $411~m$ and $385~m$, respectively, and experience high decadal retreat rates of $205~m \cdot yr^{-1}$ and $155~m \cdot yr^{-1}$. This finding agrees with a recent study of glacier retreats in the Greenland Ice Sheet[19], which identified seasonal length variability as the sole significant predictor of long-term calving front change ($R = 0.47$, Extended Data Fig. 4 in[19]) among various candidate predictors, including bed slope, surface slope, bed elevation, glacier width, terminus thickness, surface runoff, and ocean temperature. Glacier geometry is highly relevant to calving front retreats[54], especially the bed slope, as glacier can retreat faster over the overdeepened

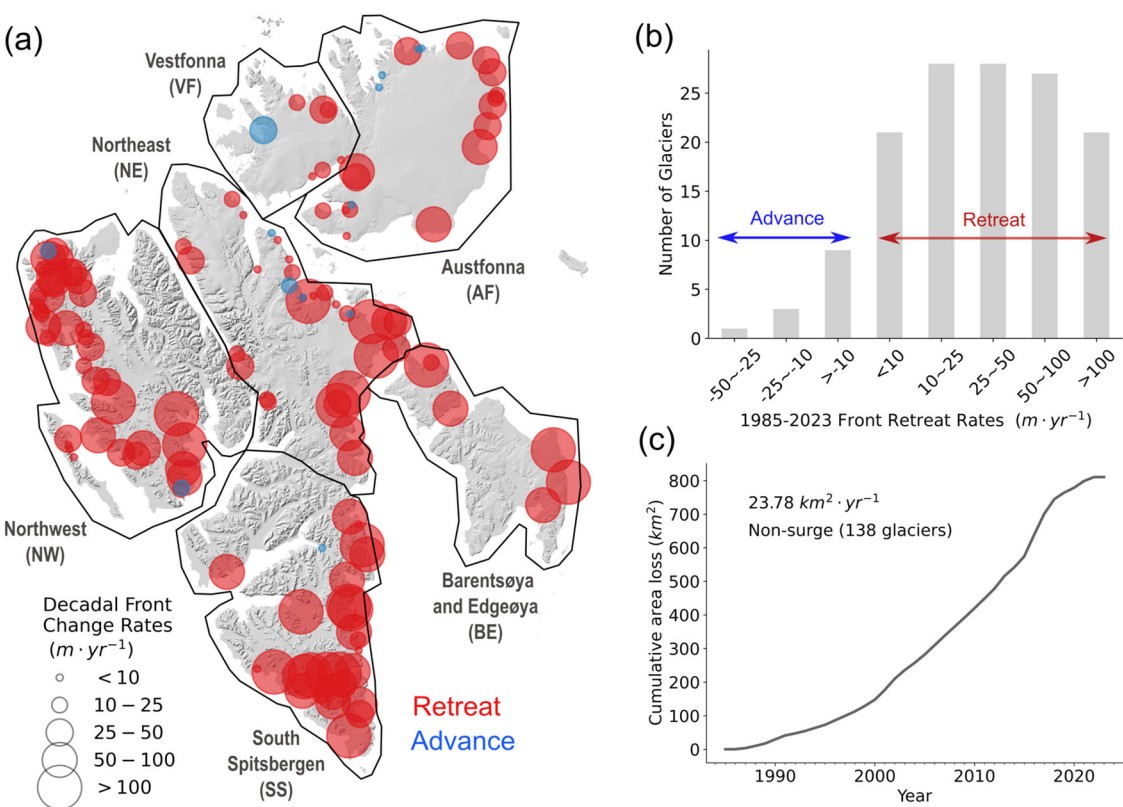

**Fig. 4 | Decadal front retreat rates. a** The decadal front retreat rates along the centreline for 138 non-surging marine-terminating glaciers in 1985-2023. Solid black polygons show the major sectors of Svalbard, including including Northeast Spitsbergen (NE), Northwest Spitsbergen (NW), South Spitsbergen (SS), Vestfonna (VF), Austfonna (AF), and Barentsøya and Edgeøya (BE). The size of the circles denotes the magnitude of decadal front change rates, red circles denote retreating glaciers while blue circles denote advancing glaciers. The background hillshade

map is generated from the 50 *m* resolution Svalbard digital elevation model (DEM) (https://data.npolar.no/dataset/dce53a47-c726-4845-85c3-a65b46fe2fea, last access: 18 April 2023). **b** The histogram of the number of glaciers in different calving front change rate categories. The blue arrow denotes advancing glaciers while the red arrow denotes retreating glaciers. **c** The cumulative calving front area loss for non-surging glaciers between 1985 and 2023 across Svalbard. Source data are provided as a Source Data file.

bed[55–57]. We found a correlation of $R = -0.53$ ($P < 0.05$) between decadal retreat rates and bed slope at calving fronts (Figure S13b), suggesting that a retrograde bed slope is associated with more significant long-term retreats. In comparison, bed slope shows no correlation with glacier retreat in the Greenland Ice Sheet[19]. The links between seasonal front range, bed slope, and decadal retreat rates indicate a glacier-specific internal dynamics are in play in controlling the rates and magnitude of calving front changes, likely to be changes in tensile stresses controlled by bed topography[19,58]. This is likely the reason that an increase in air and ocean temperature does not necessarily result in higher rates of change as the amplitude of glacier-specific front changes is influenced by complex factors, including bed topography and environmental conditions.

### Rapid glacier response to climate extremes

Positive correlations exist between interannual variability in calving front positions and interannual changes in environmental forcings (Figs. 5 and S14). Similar to seasonal front changes, the highest correlation is found for subsurface ocean temperature ($R^2 = 0.5$), followed by air temperature ($R^2 = 0.39$) and sea ice concentration ($R^2 = 0.34$), suggesting that ocean forcing is the most prominent factor in controlling the interannual calving front variability. However, air temperature also shows a comparable pattern and correlation, as well as the decreased sea ice cover and increased occurrences of sea-ice-free months around Svalbard (Figure S14). Over the past four decades, the Barents Sea has experienced a significant decline in sea ice cover, accompanied by rising ocean temperatures attributed to enhanced ocean heat transport from the West Spitsbergen Current, and strong increases in surface air temperatures over the Northern Barents Sea[41]. In response to this warming trend, the marine-terminating glaciers have been continuously retreating with positive area loss rates over the entire study period (Fig. 5). Especially between 1985 and 2016, the interannual calving front area loss rates increased in parallel with trends in air temperature and subsurface ocean temperature anomalies. Glaciers with the most significant calving front retreat rates are located along the west and southeast coast (Fig. 4a), featured by the highest increases in ocean temperature[59,60]. The effects of atmospheric and oceanic warming on long-term front retreat rates cannot be fully partitioned as they are climatically coupled[60], as noted by previous studies that emphasise the role of both factors in controlling the calving dynamics of marine-terminating glaciers in Greenland and Svalbard[21,34,39,61].

The interannual area loss rates from 1985 to 2023 capture two shifts in climate forcings in recent years (Fig. 5): 1) 2016–2018, here we refer to it as the warming phase; 2) 2019–2021, here we refer to it as the cooling phase. During the warming phase, the area loss rates across Svalbard almost doubled compared to 2015. This significant area losses are coincident with peak air and ocean temperatures in the entire 38-year record. Svalbard in 2016 is featured by extreme precipitation and air temperature and it has the wettest summer since 1955[3]. Compared to the 2010–2015 mean, air temperature in 2016 was up to 4 °C higher across Svalbard, accompanied by an up to 2 °C ocean warming (Fig. 6a). This significant temperature increase is found to be linked with a strong flux of warm Atlantic water and associated warm air driven by the Scandinavian blocking pattern[3,4]. During the cooling phase in 2019, the area loss rates decreased to the pre-2016 level (Fig. 5), accompanied by an -2 °C cooling of air temperature and up to 1 °C decrease in subsurface ocean temperature (Fig. 6b). A recent study has identified this cooling to be associated with the slowdown of a large-scale ocean circulation in the Nordic Seas due to atmospheric blocking[2]. Geodetic records of the annual mass loss for non-surging glaciers considered in this study across Svalbard shows a similar pattern over the last two decades (Figure S15a)[13]. It has been demonstrated that atmospheric blocking can significantly modify atmospheric and oceanic circulation patterns in the northern Atlantic Ocean, also

impacting submarine melting of glaciers in Greenland[2]. Compared to Greenland, Svalbard is in closer contact with the Atlantic intermediate warm water and may be impacted more directly by any slight changes in the atmospheric blocking patterns. Over the past 40 years, atmospheric blocking events have occurred more frequently and with greater intensity, primarily in response to amplified Arctic warming[4].

### Dynamic mass loss and future impact

Past studies have linked calving front retreat with increased ice dynamics in Greenland[62] and Antarctica[63]. Here we investigate the relationship between glacier front changes and mass balance components extracted from other available data products. We aggregate cumulative mass balance estimates, excluding submarine changes, from GRACE/GRACE-FO satellite gravimetry measurements using JPL Mass Concentration blocks (mascons) from Svalbard since 2002 (Figure S15b). According to these data, Svalbard has lost -368 ± 9 *Gt* of ice (1.02 *mm* sea level equivalent) over the past two decades. The rate of mass loss increased since 2012, in part caused by the massive surging of Storisstraumen in Austfonna (RGI60-07.00027)[64]. To determine the proportion of mass loss due to surface and ice dynamical processes, we calculated the cumulative surface mass balance for Svalbard land ice using the high-resolution (2.5 *km*) surface mass balance data generated from the CryoGrid model forced with CARRA (Copernicus Arctic Regional Reanalysis) data[65] (red line in Figure S15b). The mass change due to ice dynamical processes (orange line in Figure S15b) was then calculated as the difference between the total mass balance and the surface mass balance (see Methods). The results indicate that during 2002–2022, Svalbard glaciers discharged about 210 ± 13 *Gt* of ice into the ocean, accounting for 57% of the total mass imbalance. In comparison, the ice discharge in Svalbard calculated by Kochtitzky et al.[31] during 2000–2020 was 193 ± 22 *Gt*. The acceleration of the total mass loss since 2012 is caused by both surface mass balance and ice discharge, but dominated by discharge.

The sustained increase in ice discharge is likely related to our observed glacier retreat of non-surging marine-terminating glaciers, as a result of both atmospheric and oceanic warming around Svalbard. The spatial distribution of long-term calving front areal loss matches the observed long-term thinning of Svalbard glaciers in 1936–2010[11]. The current interpretation of Svalbard glacier mass loss and its future sea-level rise contribution throughout this century has focused on the role of the atmosphere (air temperature increases) and precipitation on ice mass changes[11,12,66]. Ice-ocean interaction of marine-terminating glaciers is yet to be adequately incorporated into mass loss projections of Arctic glaciers[17]. Hereby, regional climate models in recent studies have not been coupled to a dynamic ocean model[12]. As marine-terminating glaciers become exposed to warmer ocean water through Atlantification, ocean-induced undercutting in combination with surface melting is likely to increase frontal ablation, in turn leading to increased ice discharge and mass loss from Svalbard. This will at some point be counterbalanced by glaciers retreating into shallower waters or on land, but our results give no indication of that occurring yet, and about 13% of the glacier area on Svalbard is still grounded below sea level[18]. As a region highly sensitive to climate change, the behavior of Svalbard marine-terminating glaciers offers an opportunity to explore the controls on calving front changes for marine-terminating glaciers in general, which are pervasive across the Arctic archipelagos and around Greenland.

## Methods
### Glacier calving front mapping
The calving fronts of marine-terminating glaciers in Svalbard between 1985 and 2023 were mapped from optical and SAR satellite images available from Landsat, Terra-ASTER, Sentinel-1 and Sentinel-2 satellites, using a deep learning model Charting Outlines by Recurrent Adaption (COBRA)[67]. This model combines convolutional neural

networks (CNN) for feature extraction and active contour models. It outperforms the traditional image segmentation model such as U-Net in mapping glacier calving fronts. After applying the postprocessing steps developed in Li et al.[68], the final calving front dataset contains 124919 calving front traces for 149 marine-terminating glaciers in Svalbard during the past 38 years[68]. The overall calving front mapping uncertainty across Svalbard of this data product is 31 m, by measuring the mean distance error in different calving front locations mapped on the same day for a given glacier. In addition to the long time span, this calving front dataset offers unprecedented temporal density. After 2014, since the launch of the Sentinel-1 and Sentinel-2 satellites, the average temporal resolution is 4 days for all glaciers. For each glacier, the calving front change time series is generated using a centreline approach[68], by measuring the advance or retreat of the calving front in relation to their earliest position along a glacier centreline generated from the OGGM global glacier model[69].

### Time series decomposition

The detailed workflow of decomposing the seasonal calving front change cycles and long-term calving front trend from the calving front time series is shown in Figure S1, with intermediate step-by-step examples in Figures S2, S3 and S4 for three glaciers: Vestre Osbornebreen, Olsokbreen and Paierlbreen. To derive the seasonal calving front change cycle, we used the calving front time series after 2014 due to its high temporal resolution of Sentinel-1 and Sentinel-2 satellite images. We first used a 30-day median filter to remove outliers from the time series after 2014. Then we used a low-pass filter with a cutoff frequency of 180 days to detrend the filtered calving front change times series. After deriving the detrended calving front time series, the calving front seasonality was calculated as the monthly mean detrended calving front retreat by grouping the detrended time series into bins of months of observation (Figures S2, S3 and S4), the standard deviation is calculated as the uncertainty. To assess the annual self-similarity of detrended calving front change time series of each glacier after 2014, we adopted a similar approach as Wallis et al.[70] by calculating the autocorrelation at lags of one to eight years. The mean of these yearly autocorrelation values was then assigned as a single statistic over the entire nine-year time period between 2014 and 2022 for each glacier. We define an autocorrelation value ≥0.08 as strong annual periodicity, meaning this glacier displays a seasonal cycle in calving front changes.

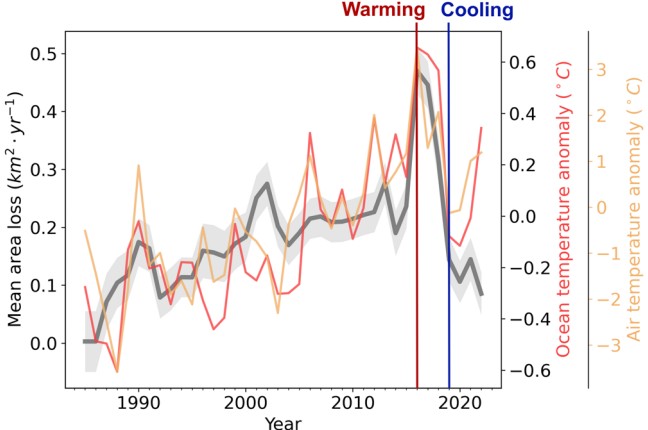

**Fig. 5 | Interannual variability in calving front area loss rates.** The mean annual area loss rate per glacier for non-surging marine-terminating glaciers across Svalbard (solid grey line) with subsurface ocean temperature anomaly (solid red line) and 3 m air temperature anomaly (solid yellow line). The vertical red line denotes significant warming in 2016 and the vertical blue line denotes subsequent cooling in 2019. The climate anomalies are calculated as the deviation of the annual mean from the 1985–2022 mean value.

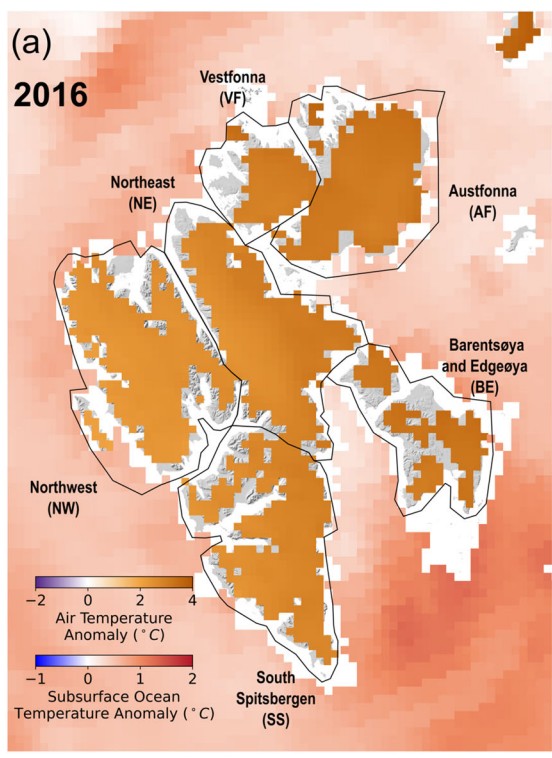

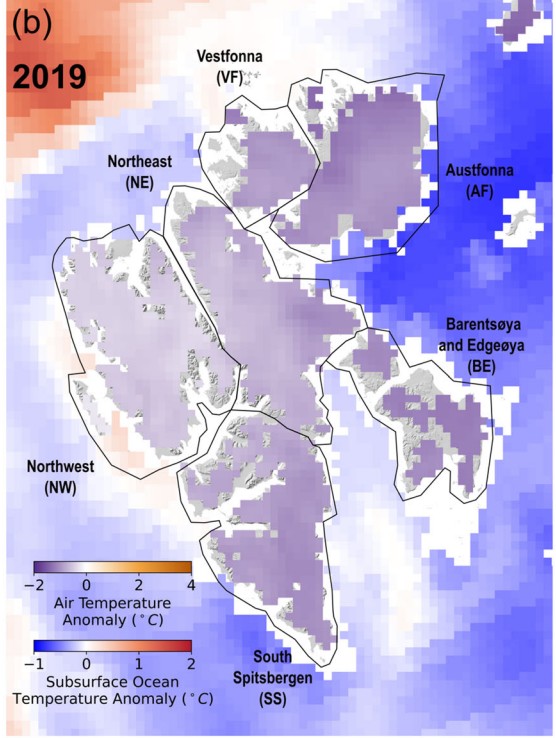

**Fig. 6 | Air and ocean temperature anomalies.** Annual mean air temperature and subsurface ocean temperature anomalies in (**a**) 2016 and (**b**) 2019 compared to the 2010–2015 mean. The background hillshade map is generated from the 50 m resolution Svalbard digital elevation model (DEM) (https://data.npolar.no/dataset/dce53a47-c726-4845-85c3-a65b46fe2fea, last access: 18 April 2023).

The interannual trend of calving front change throughout the entire study period was calculated by adding the calving front times series without the seasonal cycles after 2014 to the initial calving front time series before 2014 (Figure S1). We first calculated the annual mean calving front retreat distances along the glacier centerline by aggregating all the long-term calving front observations in bins of individual years across the observation period of a glacier. For years within this observation period that do not have any observations, we linearly interpolated the calving front change distances between neighbouring years. The calving front retreats were then converted into areal change rates by multiplying the glacier front width, which was calculated as the mean length of all terminus traces of each glacier. The uncertainty in glacier width is estimated as the standard deviation of terminus trace lengths. We assign 31 $m$[68] uncertainty to the annual calving front distance. The combined uncertainty in calving front area change is calculated as the product of the relative uncertainties of these two components. Using the mean length of terminus traces as glacier width may underestimate the area change in cases where the front traces are cut short due to restrictive front masks defined in Li et al.[68] based on Water Body Mask (WBM) and the Copernicus GLO-90 digital elevation model data, or overestimate the area change in cases where retreat along the centreline is larger than at the margins due to the typical transition from deep to shallow ocean water. However, the purpose of the area change estimation is to scale up the retreat rates by glacier size for larger-scale regional assessments, rather than providing exact measures of area change.

To evaluate the calving dynamics in different regions around Svalbard, we divided Svalbard into six different sectors based on Fig. 1 in Noël et al.[12] including Northwest Spitsbergen (NW), Northeast Spitsbergen (NE), Vestfonna (VF), Austfonna (AF), Barentsøya and Edgeøya (BE), and South Spitsbergen (SS). For seasonal calving front changes, all non-surging marine-terminating glaciers located in the same sector with a seasonal cycle (autocorrelation ≥ 0.08) were aggregated by months to calculate the mean of seasonal calving front change rates. For the sector-scale interannual calving front changes, the areal loss rates were aggregated by bins of year by calculating the average calving front areal losses for all non-surging glaciers in each sector. We estimate the uncertainty of regional retreat rates as the root mean square of annual or monthly uncertainties of all glaciers. Surging glaciers (Figure S9) were excluded from both seasonal and interannual analyses of calving front changes because surging events are not well understood and are likely driven by internal glacier dynamics[71]. The long-term cumulative calving front areal losses at the sector scale were calculated as the cumulative sum of the annual front areal losses. The excluded surging glaciers (Figure S9) were identified from the glacier calving front change time series visually if the calving front underwent significant advances > 500 $m$ over two years period[68]. These glaciers are listed in Table S2, including the Storisstraumen on Austfonna (RGI60-07.00027) which started surging in 2012[64] and still has a highly dynamic calving front.

### Bed topography and ice thickness

We derived the bed topography and ice thickness at the glacier calving front from SVIFT v1.1 - The Svalbard ice-free topography dataset[18]. It uses a mass conservation approach to reconstruct ice thickness. To calculate the mean bed topography and ice thickness for each glacier, we sampled bed topography and ice thickness within 500 $m$ of the ice margins in SVIFT dataset along the glacier centerline generated in Li et al.[68] and calculated the mean values. The bed slope is calculated by fitting a trend line to topography sampled along this 500 $m$ length centerline segment.

### MAR and ORAS5 data analysis

We derived daily surface runoff and 3-m air temperature from 1985 to 2023 from a Regional Climate Model (RCM) Modèle Atmosphérique Régional (MAR)[72,73] with 6 km spatial resolution. Data for our calving front analysis were extracted from a location upstream of the calving front traces time series of each studied glacier. The glacier point was firstly converted from the centre point of each glacier domain polygon defined in Li et al.[68], it was then manually adjusted based on the MAR ice mask file and the final glacier calving front traces time series. The climate variables from MAR daily files were interpolated at each glacier point using the nearest neighbour interpolation. The spatial distributions of the glacier points are shown in Figure S5.

Subsurface ocean temperatures were derived from the ECMWF ORAS5 (Ocean Reanalysis System 5) global ocean reanalysis monthly dataset from 1985 to 2023[74]. The horizontal resolution of the reanalysis data is ~0.15° in Svalbard and we gridded data into 0.15° regular grids. This spatial resolution is enough to capture the spatiotemporal variability of ocean temperature outside the fjords as the width of the fjord mouth is normally larger than the grid size of ORAS5 data. The subsurface ocean temperature calculated as the mean temperature across the 20–100 $m$ water depth, was derived at the fjord point of each studied glacier across Svalbard. The fjord point was derived by finding the closest non-empty grid point in the ORAS5 sea surface temperature grid to the corresponding glacier point of each glacier derived in the MAR data processing. The spatial distribution of the fjord points is shown in Figure S6.

### Sea ice concentration

Sea ice concentration represents the areal fractional coverage of sea ice. To investigate the relationship between sea ice concentration and glacier calving front changes, we generated the sea ice concentration time series over the time period of 1985–2023 for every glacier studied. We use three different data sources including: 1) sea ice concentrations from Nimbus-7 SMMR and DMSP SSM/I-SSMIS Passive Microwave Data at a spatial resolution of 25 $km$ with a temporal coverage from 1985–2002[75]; 2) sea ice concentrations from AMSR-E (Advanced Microwave Scanning Radiometer for EOS) at a spatial resolution of 3.125 $km$ during 2002–2011[76]; 3) sea ice concentrations from AMSR-2 from 2012 to 2023 at a spatial resolution of 3.125 $km$[77]. The daily sea ice concentration was derived at the fjord point of each studied glacier across Svalbard. When summarising the sea-ice-free years in each month in the seasonality analysis (Figures S7 and S8), the sea-ice-free year is defined as when the mean sea ice concentration is less than 20%. To determine the relationship between glacier calving front changes and time-evolving environmental variables including surface runoff, air temperature, subsurface ocean temperature, and sea ice concentration, linear regressions between parameters are applied in the correlation analysis at both seasonal and interannual timescales. In the sector-scale long-term calving front changes and their responses to climate forcings, we calculated the long-term climate anomalies by using the 1985–2022 mean as the baseline.

### Mass balance data processing

We used the GRACE JPL Mass Concentration blocks (mascons) over Svalbard to estimate the cumulative mass balance from 2002 to 2022 at monthly temporal resolution[78,79]. The JPL mascons fields are based on the Level-1B GRACE/GRACE-FO data, its C20 and C30 coefficients are replaced with solutions from satellite laser ranging in Technical Note TN-1466 and the degree-1 coefficients (trend) are estimated using methods from Sun et al.[80]. In addition, the JPL mascons have been corrected for the glacial isostatic adjustment based on the ICE6G-D model[81]. We fit a 12-month low-pass filter to the monthly GRACE mass change time series to derive the interannual trend, and the root mean square error (RMSE) between this filtered long-term trend and the original mass change time series is estimated to be the uncertainty of total mass change. For the surface mass balance over the same period as the JPL mascons, we used the monthly high-resolution (2.5 $km$) Svalbard glacier mass balance simulations generated from the

CryoGrid community model forced by Copernicus Arctic Regional ReAnalysis (CARRA) data[65]. The same low-pass filter was applied to the monthly SMB time series for the interannual trend, and its uncertainty was also calculated as the RMSE between the interannual SMB trend and the monthly time series. The mass change due to ice discharge, ignoring calving front position changes, was then calculated as the difference between the total mass balance and the surface mass balance (Figure S15b), and its uncertainty was estimated as the root mean square of the uncertainties from these two different components.

To separate mass changes from surging and non-surging glaciers, we used the per-glacier annual mass change time series from Hugonnet et al.[13] derived from differencing of ASTER DEMs from 2000 to 2020. Based on the marine-terminating glacier calving front data product generated in our study, we updated the RGI6.0 glacier database to address the mismatch in glacier types between Hugonnet et al.[13] dataset and our glacier calving font product[68,82]. We then generated an annual time series of mass loss for non-surging glaciers (Figure S15a). Excluded surging glaciers are listed in Table S2.

## Data availability
Glacier calving front traces and front change time series used in this article can be downloaded at https://zenodo.org/records/10407266. The Regional Climate Model Modèle Atmosphérique Régional (MAR) data for Svalbard is available upon request from Xavier Fettweis. The ORAS5 ocean reanalysis data is available at https://cds.climate.copernicus.eu/datasets/reanalysis-oras5?tab=overview. SMMR, SSM/I-SSMIS sea ice concentration dataset is available at https://nsidc.org/data/nsidc-0051/versions/2. AMSR-E and AMSR-2 sea ice concentration datasets are available from University of Bremen at https://seaice.uni-bremen.de/data-archive/. The GRACE JPL Mass Concentration blocks (mascons) can be downloaded at https://podaac.jpl.nasa.gov/dataset/TELLUS_GRAC-GRFO_MASCON_CRI_GRID_RL06.1_V3#. The CryoGrid CARRA surface mass balance data over Svalbard can be downloaded at https://adc.met.no/datasets/10.21343/ncwc-s086. The SVIFT Svalbard bed topography and ice thickness can be downloaded at https://data.npolar.no/dataset/57fd0db4-afbf-4c94-ac1c-191c714f1224. Source data are provided with this paper.

## Code availability
The source code of COBRA model v1.0.0 for calving front mapping and inference examples are accessible at https://github.com/khdlr/COBRA/releases/tag/v1.0.0 (last access: 10 January 2023), its DOI is https://doi.org/10.5281/zenodo.8407566.

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

## Acknowledgements

This work is funded by the European Union's Horizon 2020 research and innovation programme through the project Arctic PASSION (grant number: 101003472). J.B. also received funding from the German Federal Ministry of Education and Research (BMBF) in the framework of the international future AI lab "AI4EO – Artificial Intelligence for Earth Observation: Reasoning, Uncertainties, Ethics and Beyond" (grant number: 01DD20001). K.H. received funding from the German Federal Ministry for Economic Affairs and Climate Action in the framework of the "national centre of excellence ML4Earth" (grant number: 50EE2201C). X.X.Z. received funding from the Munich Centre for Machine Learning (MCML). We would like to thank Xavier Fettweis from the University of Liège for providing MAR data, NSIDC for the SMMR and SSM/I-SSMIS sea ice concentration data, the University of Bremen for AMSR-E and AMSR-2 sea ice concentration data, NASA JPL for the GRACE JPL Mass Concentration blocks data, and the Norwegian Meteorological Institute for the CryoGrid CARRA surface mass balance data and the SVIFT (the Svalbard ice-free topography) data.

## Author contributions

T.L. and J.B. conceived the study. T.L. designed the research, conducted the analysis, and wrote the manuscript. S.H. contributed to the processing of MAR data and analysis of the results. G.M. and J.B. contributed significantly to improving the analysis and interpretation. A.I., K.H., and X.X.Z. contributed to time series decomposition and statistical analysis. T.L., S.H., G.M., A.I., K.H., X.X.Z., and J.B. contributed to the writing of the manuscript.

## Funding

## Competing interests

The authors declare no competing interests.
