## [Transparent Peer Review file · Nature Communications]

Pervasive glacier retreats across Svalbard from 1985 to 2023

Corresponding Author: Dr Tian Li

Version 0:

Reviewer comments:

Reviewer #1

(Remarks to the Author)

The paper "Atmospheric and oceanic warming drives pervasive retreat of marine terminating glaciers in Svalbard" deals with the delineation of glacier front positions of 149 marine-terminating glaciers in Svalbard. They use these front positions to describe decadal and seasonal variations and find a correlation with atmospheric and oceanic forcings, a correlation of decadal and seasonal variations and highlight the importance of ice-ocean processes at the calving front. The 124919 glacier front positions, that are created by the authors represent a novel, extensive and interesting dataset. However, we highly question the novelty of their findings regarding the analysis for reasons explained below.

The fact that the glaciers in Svalbard are governed by atmospheric and oceanic forcing is not a novelty, as the authors themselves show in the introduction (lines 32-42), where they provide several examples of studies showing that. Also, the importance of incorporating ice-ocean interactions has been well-established (Straneo and Heimbach, 2013; Cowton et al., 2018; Catania et al., 2020). Therefore, in our opinion the findings do not add new insights to the existing literature.

Furthermore, a main finding of the paper is the "significant" positive correlation between seasonal and decadal front positions (lines 77-83). However, we question the significance of this finding, which is specified with an R^2 of 0.35. The authors compare this number to a study from Greenland (Greene et al. 2024) where an R^2 of 0.71 was found. Instead of drawing the same conclusions as for Greenland, the authors should investigate why it is more challenging to observe a positive correlation for Svalbard glaciers compared to Greenland. Factors such as glacier size, velocity, and fjord depth could be explored.

Other major comments:

- Calving front positions are interpreted without taking flow velocities into account, which could change the observed pattern of calving immensely.
- Line 17-19: It states that glaciers with seasonal calving cycles are more sensitive to long-term climate change. This requires further investigation into glacier front geometry for a representative sample of glaciers with large and small seasonality. They should explore why some glaciers exhibit seasonal cycles while others do not, and whether this could be related to glacier geometry and potential changes as glaciers may retreat to new pinning points.
- Line 65: The uncertainty of the annual areal change rate of $0.47 \pm 1.01 \text{ km}^2/\text{yr}$, seems very high to us, being more than twice as large as the amount of the actual value mentioned. Also, the mean seasonal rate of $91.3 \pm 75 \text{ m}$ (line 75) has very large uncertainties, making it difficult to rely on these results.
- Line 75: There are inconsistencies between the text and figures. For instance, line 75 refers to Olsokbreen showing a 411 m seasonal front range, which is double the range shown in figure 2a.
- Line 79-80: The wording of the two examples given can be misleading. Potentially focusing on one example or breaking the sentence apart leads to more clarity.
- Line 105, 153: On several occasions the authors treat calving as equivalent to front position changes, therefore neglecting submarine melt and changes in ice flow velocity.
- Line 119-120: The argument is invalid. Holland et al. (2008) suggested that changes in Jakobshavn after 1997 were triggered by ocean warming but did not state that the peak of 15 m/yr was a result of a 2-degree ocean warming. This needs to be corrected as well as their concluding remarks based on this.
- Line 20 states: " There is also a strong correlation between grounded ice mass loss and front retreat, indicating that calving can be used as a proxy for the long-term mass loss under climate change." but, the manuscript does not explain grounded ice mass loss or discuss the use of calving as a proxy for long-term mass loss, and also the role of submarine melt is missing.
- Line 82: The argument for extrapolating retreat rates to longer time scales is not convincing, as retreat rates also depend on the glacier front geometry. As glaciers retreat, they may reach new pinning points that could reduce seasonal sensitivity.

This needs a more detailed explanation, especially considering the R2 value of 0.35.

- Line 142: "In addition, the peaks in annual cumulative surface runoff anomaly coincide with the peaks in mean interannual areal loss rates (Figure 4a), indicating surface runoff has an instant effect on calving front retreat rates." Figure 4, there are other anomaly peaks coinciding with the peak in areal loss (panel b & c). The text should clearly identify why they only use panel a to explain what affects calving front retreat rates.
- Line 172-175: The logic in this part is unclear and needs to be elaborated.
- Line 195-200: The differences between NW and NE Svalbard and the influence of atmospheric and oceanic temperatures on each other are not clearly explained and need further detail.
- Line 201-202: "In addition, our study finds that the drastic ice speed change ranging from surging events does not impact the seasonal calving cycles (Figure S2)." this part does not seem to be linked to the rest of the manuscript.
- Surging glaciers were mentioned a few times in the manuscript, but it does not explain the differences between surging and non-surging glaciers, or why the distinction is important. This needs to be clarified.
- Line 238-242: Again, this part does not seem to be linked to the rest of the manuscript. It should be integrated better or clarified.
- Figure captions in the supplementary materials are incomplete, making it difficult to follow their arguments. The figures themselves are not self-explanatory and would benefit from more detailed descriptions. For example, figure 1c doesn't show errors, figure 2a doesn't say what the size of squares means, figures S2,S3 in panel f what is dotted line?

Reviewer #2

(Remarks to the Author)

Reviewer #3

(Remarks to the Author)

The paper assesses controls on glacier retreat in Svalbard. Thus, it tackles an interesting topic and has a wealth of data, which can be used to investigate some really important questions. Overall, the paper presents a number of novel and interesting results, e.g. seasonal patterns on Svalbard, the correlation between season changes and inter annual and that surge type glaciers seem to have a similar pattern as non-surge type (although the final point is noted but not properly explored). However, the paper feels like it needs a bit more thought and precision, to ensure it really does disentangle these important questions. There are several examples of this, like the surge seasonality point, whether surface elevation / SMB changes are caused by retreat or cause it (i.e. the direction of the relationship, whether the relationship between total mass loss and area change is really pervasive given the apparent impact of two data points on the relationship, whether sea ice plays any role (it is mentioned as an explanation a few times, but never really explored properly). There are also small scale examples of this, e.g. melt rates are taken from Jakobshavn Isbrae but there isn't much consideration of how appropriate these might be, using frontal ablation and calving interchangeably, adding / removing the surge type glaciers in different parts, making it confusing about when they are in/out. I know that Nature Comms is short format, which makes exploring ideas difficult, but unfortunately some of the arguments did not feel fully pinned down and explored, which makes the narrative less convincing. I think you have a really solid dataset here and some really interesting findings, so regardless of what is decided by Nature Comms, I strongly encourage submission elsewhere and this might give you the space to fix some of the issues noted above. Please see my annotated pdf for detailed comments.

Reviewer #4

(Remarks to the Author)

Review of Atmospheric and oceanic warming drives pervasive retreat of marine-terminating glaciers in Svalbard by Li et al.

General Comments

This is an interesting article which relates environmental variables to glacier retreat on a pan-Svalbard scale. Overall, I think the manuscript is well written and scientifically sound. However, I have one general comment related to the location of the fjord points used for extracting the oceanographic data. In the supplementary material, Fig S5 shows the fjord points used. Many of these fjord points are a considerable distance from the glacier fronts, and even outside of the fjord mouth. Previous research has shown that the water masses present in the inner fjords can be quite different to those present outside of the fjord mouth (see e.g. Fig. 4 in Prominska et al, <https://doi.org/10.1016/j.oceano.2017.07.003> and/or Divya et al <https://doi.org/10.1016/j.polar.2021.100735>). I understand that the location of these fjord points may be an unavoidable consequence of the pan-Svalbard scale of the study / the datasets used, but feel that this needs to be addressed in the main text. For example, a short paragraph could be added to the discussion.

In addition, I have several minor comments which are listed below:

Minor comments

L38: I would appreciate a little more context here: is frontal ablation not considered in other studies, or assumed to be included in some other metric etc?

L60: 'Area at the calving front' – I see this explained in the methods, but was not fully sure what this meant until I found this

explanation. Maybe a short explanation can be included in the text here

L64: I appreciate the comparison with Kochitsky et al, as well as the possible explanation for the difference between their results and yours. I wonder if you would get a more similar result if you used lower (temporal) resolution data / have you tried this and was this the outcome? Then it would be clear if this was the reason for the difference in results, or suggest that something else is at play

L78: It is not clear to me whether your description of the R2 values as significant means you think this shows a strong relationship, or whether you just mean the result is statistically significant. Related to this, it would be useful if you state what you consider a 'significant' R2 value to be, and also what the p values are for each R2 value. A value of 0.37 is perhaps quite good for a natural system, but is much lower than any of your R2 values linking retreat rates to environmental values, so some discussion of how you interpret the R2 values would be beneficial.

L83: Do you have any suggestions for why the seasonal range/long-term range R2 value is twice as high for Greenland?

L113: Interesting that the r2 value for subsurface ocean temperatures is very similar to that for one month lagged air temperatures – do the time series for these two variables look similar?

L120: Please state whether this is basal melt under grounded or floating ice, and also specify if this is an additional 15 m/yr of basal melting or an absolute value.

L124: You could link this to e.g. Slater and Straneo (<https://www.nature.com/articles/s41561-022-01035-9>), where atmospheric temps was found to exert a first order control on submarine melt

Fig. 3: Please add p values to the annotation on the plots (the same for other figures)

L175: Could it also be some internal dynamic feedbacks? E.g. climate drivers cause front retreat, which cause acceleration and then thinning etc?

L204: I think this should be 'mirroring a recent finding FROM/FOR THE Greenland Ice Sheet'

L205: Reference should be in brackets

L318: Can you add a comment on the impact of the resolution of the ocean data set, given the known importance of fjord circulation for submarine melt rates

Version 1:

Reviewer comments:

Reviewer #3

(Remarks to the Author)

I feel that the authors have provided a comprehensive, effective and well-considered response to my comments. I am satisfied by their responses and think they have a really nice paper with an excellent dataset, that provides novel insights into Arctic glacier behaviour. They should be commended on a really nice piece of work and their throughout response to reviewer comments. I have no further comments and look forward to seeing the final version.

Reviewer #4

(Remarks to the Author)

Thank you to the authors for your comprehensive responses to all the reviews, and the associated changes in the manuscript. The edits have, in my opinion, led to significant improvements and I do not have any major concerns with the article in its current form.

However, I would ideally like to see your statement regarding the ocean temperature dataset changed. I appreciate you adding in more information about the dataset but am not fully convinced by the statement that the distance of data points from the coast 'does not impact the relationship between ocean warming and seasonal calving front retreats'. Data/figures from e.g. Promińska et al that show AW in your chosen depth range is present in the outer fjord but not near the glacier front in several years (e.g. 2002, 2007, 2013, 2014). Other previous research has showed that the relationships between ocean temps and frontal ablation is different when you take ocean data from closer to the calving front (Holmes et al.). Whilst I am onboard with your approach being appropriate given the scale of the study, the statement quoted above needs to be toned down.

One minor comment is that your units are sometimes italicized (e.g. L164) but sometimes not (e.g. L169) – this should be consistent throughout the manuscript.

Responses to Reviewers' Comments

Tian Li on behalf of other co-authors

Reviewer 1

The paper “Atmospheric and oceanic warming drives pervasive retreat of marine terminating glaciers in Svalbard” deals with the delineation of glacier front positions of 149 marine-terminating glaciers in Svalbard. They use these front positions to describe decadal and seasonal variations and find a correlation with atmospheric and oceanic forcings, a correlation of decadal and seasonal variations and highlight the importance of ice-ocean processes at the calving front. The 124919 glacier front positions, that are created by the authors represent a novel, extensive and interesting dataset. However, we highly question the novelty of their findings regarding the analysis for reasons explained below.

The fact that the glaciers in Svalbard are governed by atmospheric and oceanic forcing is not a novelty, as the authors themselves show in the introduction (lines 32-42), where they provide several examples of studies showing that. Also, the importance of incorporating ice-ocean interactions has been well-established (Straneo and Heimbach, 2013; Cowton et al., 2018; Catania et al., 2020). Therefore, in our opinion the findings do not add new insights to the existing literature.

Thank you very much for your constructive and detailed comments, they have significantly improved the quality of this research.

First, in the initial submission, we claim that the influence of atmospheric and oceanic forcing on the retreats of marine-terminating glaciers is well established in the Greenland Ice Sheet, not in Svalbard, due to a lack of dense calving front measurements that cover a long time span and large spatial coverage. Since the Arctic has diverse oceanic, atmospheric and glaciological settings (please see comments from Reviewers 2&3), what has been found for Greenland may not apply to Svalbard. So far, the air-ice-ocean interactions in Svalbard have only been investigated for a few marine-terminating glaciers and over a short time period. Therefore, climate drivers and the governing processes for calving front dynamics remain largely unknown across Svalbard. Our research now fills this knowledge gap by conducting the first comprehensive analysis of seasonal and long-term calving front changes and their relationships with different environmental variables across the entire Svalbard over the past 38 years. This is a novelty.

Second, the current mass balance measurements for Svalbard have not incorporated the frontal ablation of marine-terminating glaciers [1, 2] and the mass loss projections of most glacier models did not include this component [3], therefore we demonstrate that it is crucial to examine the mechanism of marine-terminating glacier retreats, especially in Svalbard, a region located in the most rapidly warming region of the Arctic.

Third, in revision, we have implemented additional analyses based on the review com-

ments to investigate spatiotemporal variabilities in calving front changes at different timescales and have significantly restructured the paper. The key messages and results have been greatly improved compared to the initial submission.

Here we briefly outline the main novelties in the revision:

- We find that that 91% of the non-surge marine-terminating glaciers have been retreating since 1985 and 62% of them have seasonal cycles across Svalbard. No study before has done such a detailed analysis over a long period of almost 40 years. In addition, previous studies on seasonal calving front change cycles have only focused on specific glaciers or small regions. This conclusion has provided new understanding into the historical changes and current status of marine-terminating glaciers in Svalbard and allows us to compare with other ice masses in the Arctic such as the Greenland Ice Sheet.
- From western to eastern Svalbard, there are several months' time delays in seasonal calving front retreat rates, coincident with the regional variability in ocean warming, demonstrating the dominant role of ocean forcing on seasonal calving front cycles. We are the first to directly observe such a spatial pattern in calving fronts across Svalbard, and demonstrate the clear link between warm water inflow and the timing in seasonal calving front changes.
- We find the interannual variability in calving front rates is sensitive to both atmospheric and ocean warming. Previous studies on frontal ablation of marine-terminating glaciers in Svalbard mainly focused on ocean processes, instead of both. This further supports the conclusion by [4] on the Greenland Ice Sheet, highlighting that long-term dynamic mass loss cannot be adequately addressed by focusing solely on ice-ocean interactions without also considering the effects of atmospheric warming.
- We find immediate responses of glacier calving front changes to large air and ocean temperature anomalies in 2016 and 2019. The large temperature anomalies during 2016-2020 have been flagged as climate extremes by two recent studies [5, 6], likely driven by atmospheric blockings in the Arctic. The possible link between atmospheric blockings and changes in marine-terminating glaciers has never been discovered before for Svalbard.

In summary, our study provides significant advances in understanding the seasonal cycles and long-term retreat of calving fronts in Svalbard, through a high-resolution calving front dataset with exceptional spatial and temporal coverage. These findings enhance our knowledge of calving processes in Svalbard and can contribute to improving the parameterization of calving in glacier models.

Furthermore, a main finding of the paper is the “significant” positive correlation between seasonal and decadal front positions (lines 77-83). However, we question the significance of this finding, which is specified with an R^2 of 0.35. The authors compare

this number to a study from Greenland (Greene et al. 2024) where an R^2 of 0.71 was found. Instead of drawing the same conclusions as for Greenland, the authors should investigate why it is more challenging to observe a positive correlation for Svalbard glaciers compared to Greenland. Factors such as glacier size, velocity, and fjord depth could be explored.

Thank you for this suggestion which is very helpful.

We would like to admit that our previous conclusion regarding the correlation between these two variables in Svalbard being lower than in Greenland Ice Sheet was incorrect. In fact, the correlations are very similar. The R^2 value of 0.71 for the Greenland Ice Sheet in [7] was calculated between the long-term mass loss and seasonal mass loss, not the effective length changes that we measured in Svalbard. In the **Extended Data Fig. 4** in [7], they normalized the terminus mass change values by terminus area and calculated the correlation between seasonal range and long-term changes in length variability, the **R-value** is only 0.47, which is slightly lower than the R-value of 0.6 ($P < 0.05$) for Svalbard (Figure S13a). Therefore, we confirm that the relationship is nearly identical between Svalbard and Greenland. We have corrected this statement in Line 169 as 'This finding agrees with a recent study of glacier retreats in the Greenland Ice Sheet'.

In addition, we also found a negative correlation ($R = -0.53$, $P < 0.05$) between bed slope and decadal retreat rates in Svalbard (Figure S13b), indicating that glaciers retreat faster along a retrograde bed slope, a theory that has been well established for marine-terminating glaciers and ice shelves [8, 9, 10, 11]. In comparison, [7] did not observe a similar relationship, it is currently unknown why marine-terminating glaciers in Greenland do not display similar changes along a retrograde bed slope in [7]. Nonetheless, our main focus is Svalbard, any investigation on the Greenland Ice Sheet is out of the research scope of this study.

Calving front positions are interpreted without taking flow velocities into account, which could change the observed pattern of calving immensely.

Thank you for pointing this out. First, here we are observing calving front changes, we consider calving front advance as when ice discharge exceeds frontal ablation and retreat as when frontal ablation exceeds ice discharge. We are not able to quantify ice discharge, and therefore frontal ablation, due to a lack of ice velocity dataset of sufficient spatial coverage and temporal resolution for the entire Svalbard over a long period that is required for this study. However, calving front retreat rate has proved to be a good indicator for the variability of frontal ablation on Svalbard [12, 13]. In these studies where ice discharge has been taken into account, it is shown that frontal ablation in winter almost shut down, allowing most of the ice discharge to go into the seasonal readvance of the calving fronts. This has now been clarified in Line 70-77.

Line 17-19: It states that glaciers with seasonal calving cycles are more sensitive to long-term climate change. This requires further investigation into glacier front geom-

etry for a representative sample of glaciers with large and small seasonality. They should explore why some glaciers exhibit seasonal cycles while others do not, and whether this could be related to glacier geometry and potential changes as glaciers may retreat to new pinning points.

Thank you for this comment. We have now removed the statement on correlation between seasonal front changes and long-term front change rates from the abstract and no longer listed this discovery as a main finding. In the revision, we set a threshold of calving front length change annual periodicity autocorrelation value ≥ 0.08 to define a glacier having seasonal cycles, instead of using ≥ 0.1 in the initial submission to classify strong and weak seasonal cycles. The reason for this change is because for some glaciers, a low autocorrelation value does not necessarily indicate weak seasonal cycles, instead, it reflects a high noise-to-signal ratio in the detrended time series, which makes it difficult to detect any meaningful seasonal changes. To make it simple, we set a uniform threshold of 0.08, this is based on the visual checks of the detrended calving front change time series in 2014-2023 of all studied glaciers.

We investigated the correlations between seasonal front range and six different potential predictors for seasonality shown in Figure R1 below, including bed slope, bed elevation, ice thickness, glacier width, air temperature seasonality and subsurface ocean temperature seasonality - calculated as the standard deviation of monthly mean temperatures. We found no clear correlations with all of these variables, indicating that the amplitudes of seasonal changes are likely caused by glacier-specific internal dynamics.

Line 65: The uncertainty of the annual areal change rate of 0.47 ± 1.01 km²/yr, seems very high to us, being more than twice as large as the amount of the actual value mentioned. Also, the mean seasonal rate of 91.3 ± 75 m (line 75) has very large uncertainties, making it difficult to rely on these results.

Thank you for pointing this out. We made a mistake in calculating the errors for both interannual area change rates and seasonal front retreat rates in each region by calculating the standard deviation of these two metrics among all different glaciers. Now we have corrected this by using the error propagation principles (please see Online Methods section 'Time series decomposition'), and all the related figures and tables have been updated to reflect the newly calculated uncertainties. At the Svalbard scale, the updated mean annual areal change rate in 2016 is now 0.47 ± 0.04 km²/yr (Figure 5), and the mean seasonal range for 86 non-surge marine-terminating glaciers is 102 ± 6 m.

Line 75: There are inconsistencies between the text and figures. For instance, line 75 refers to Olsokbreen showing a 411 m seasonal front range, which is double the range shown in figure 2a.

Thank you for the comment. The figure in the initial submission was plotted correctly; however, the mismatch arose from the arbitrary thresholds we manually set for the

Figure R1: Seasonal calving front range for 86 non-surging glaciers with seasonal cycles and their linear correlations with six predictors of sensitivity to climate change, including bed slope (a), bed elevation (b), ice thickness (c), glacier width (d), air temperature seasonality (e), and subsurface ocean temperature seasonality (f).

minimum and maximum values of the error bar to enhance visualization. We have now updated this in Figure 1a.

Line 79-80: The wording of the two examples given can be misleading. Potentially focusing on one example or breaking the sentence apart leads to more clarity.

Thank you and agree, we have now changed the wording to 'Glaciers Olsokbreen (Figure S3) and Paierlbreen (Figure S4) have high seasonal ranges of 411 m and 385 m, respectively, and experience high decadal retreat rates of 205 m/yr and 155 m/yr.' in Line 167-169.

Line 105, 153: On several occasions the authors treat calving as equivalent to front position changes, therefore neglecting submarine melt and changes in ice flow velocity.

Thank you for this comment, we have now made it clear that we observe 'calving front change' in this study throughout the manuscript, please also see our previous

response.

Line 119-120: The argument is invalid. Holland et al. (2008) suggested that changes in Jakobshavn after 1997 were triggered by ocean warming but did not state that the peak of 15 m/yr was a result of a 2-degree ocean warming. This needs to be corrected as well as their concluding remarks based on this.

Agree, we have now removed this statement from the manuscript.

Line 20 states: " There is also a strong correlation between grounded ice mass loss and front retreat, indicating that calving can be used as a proxy for the long-term mass loss under climate change." but, the manuscript does not explain grounded ice mass loss or discuss the use of calving as a proxy for long-term mass loss, and also the role of submarine melt is missing.

Thank you for this comment and agree. We have removed the correlation analysis between grounded ice mass loss rates with front retreat rates from the paper, as Reviewers 2 and 3 pointed out that this correlation is severely skewed by two data points which makes this interpretation less reliable.

Line 82: The argument for extrapolating retreat rates to longer time scales is not convincing, as retreat rates also depend on the glacier front geometry. As glaciers retreat, they may reach new pinning points that could reduce seasonal sensitivity. This needs a more detailed explanation, especially considering the R2 value of 0.35.

This is a very good point, especially considering that our latest analysis on possible predictors for the long-term calving front retreat rates in Figure S13 shows that there is a negative correlation between decadal retreat rates and bed slope at calving fronts ($R=-0.53$, $P<0.05$), indicating that a retrograde bed slope is likely contributing to higher long-term retreat rates. We have now removed the statement on extrapolating retreat rates to longer time scales.

Line 142: "In addition, the peaks in annual cumulative surface runoff anomaly coincide with the peaks in mean interannual areal loss rates (Figure 4a), indicating surface runoff has an instant effect on calving front retreat rates." Figure 4, there are other anomaly peaks coinciding with the peak in areal loss (panel b and c). The text should clearly identify why they only use panel a to explain what affects calving front retreat rates.

Thank you for this suggestion. We agree and have significantly changed the interpretation of results related to the interannual variability in calving front change rates, now in Section 'Rapid disintegration in response to climate extremes'. We now explain the air-ice-ocean interactions mainly based on correlation analysis in Figure S14 and Table S4. In addition, we found that there is an immediate response of glacier front changes to high air and ocean temperature anomalies in 2016 and 2019, which are possibly caused by atmospheric blockings according to [6, 5].

Line 172-175: The logic in this part is unclear and needs to be elaborated.

Thank you for this comment, we have now removed the entire paragraph from the manuscript as we no longer correlate calving front area loss rates with Hugonnet et al. (2022) mass loss rates. Please also see our responses to Reviewer 3.

Line 195-200: The differences between NW and NE Svalbard and the influence of atmospheric and oceanic temperatures on each other are not clearly explained and need further detail.

Thank you for this comment. We have plotted the regional seasonal front retreat rates and monthly mean air and subsurface ocean temperatures for six different sectors in Figure 3. Examples of detrended front change time series of typical glaciers with different environmental variables are plotted in Figure 2. The results and associated explanation are now in Section 'Ocean temperature variability drives divergent seasonal cycles'. From these two figures, we see the divergence in the timings between peak air temperature and peak subsurface ocean temperature grows larger from western to eastern Svalbard. While the peak air temperature remains fixed in July across different regions, the changes in the timing of peak seasonal retreat rates match the peak subsurface ocean temperature.

Line 201-202: "In addition, our study finds that the drastic ice speed change ranging from surging events does not impact the seasonal calving cycles (Figure S2)." this part does not seem to be linked to the rest of the manuscript. Surging glaciers were mentioned a few times in the manuscript, but it does not explain the differences between surging and non-surging glaciers, or why the distinction is important. This needs to be clarified.

Good point, we have now made it clear how we defined surging-type glaciers in the Online Methods Line 329-331: 'The excluded surging glaciers were identified from the glacier calving front change time series visually if the calving front underwent significant advances > 500 m over two years period.' In addition, we analyzed the seasonal cycles of surge-type glaciers and their relationship with different environmental variables including sea ice concentration in Figures S9 and S10. We then compared the seasonality in the calving front changes between non-surging and surging glaciers. These contents can be found in the Section 'Widespread seasonal calving front changes across Svalbard'.

Line 238-242: Again, this part does not seem to be linked to the rest of the manuscript. It should be integrated better or clarified.

Thank you for this comment and agree. We have removed Line 238-239 in the revision but still kept Line 240-242 as a concluding remark for the paper.

Figure captions in the supplementary materials are incomplete, making it difficult to follow their arguments. The figures themselves are not self-explanatory and would

benefit from more detailed descriptions. For example, figure 1c doesn't show errors, figure 2a doesn't say what the size of squares means, figures S2,S3 in panel f what is dotted line?

Thank you for this comment, we have now revised the supplementary figures and added detailed information in the caption for all the figures.

Reviewer 2

Reviewer 3

The paper assesses controls on glacier retreat in Svalbard. Thus, it tackles an interesting topic and has a wealth of data, which can be used to investigate some really important questions. Overall, the paper presents a number of novel and interesting results, e.g. seasonal patterns on Svalbard, the correlation between season changes and inter annual and that surge type glaciers seem to have a similar pattern as non-surge type (although the final point is noted but not properly explored).

Thank you very much for your time in providing constructive and detailed comments on this study which have been very useful in improving the analysis and results in the revision.

However, the paper feels like it needs a bit more thought and precision, to ensure it really does disentangle these important questions. There are several examples of this, like the surge seasonality point, whether surface elevation / SMB changes are caused by retreat or cause it (i.e. the direction of the relationship, whether the relationship between total mass loss and area change is really pervasive given the apparent impact of two data points on the relationship, whether sea ice plays any role (it is mentioned as an explanation a few times, but never really explored properly). There are also small scale examples of this, e.g. melt rates are taken from Jakobshavn Isbrae but there isn't much consideration of how appropriate these might be, using frontal ablation and calving interchangeably, adding / removing the surge type glaciers in different parts, making it confusing about when they are in/out.

Thank you for the comments and agree. In the revision, we have added additional experiments based on the review comments and significantly changed the paper structure and narration. Major changes are outlined below:

- We added regional analysis for seasonal calving front change variability and its relationship with seasonal atmospheric and oceanic warmings.

- We excluded all the surging glaciers when calculating the regional seasonal and interannual retreat rates.
- We analyzed the seasonal calving front cycles for surging glaciers and compared them with non-surging glaciers.
- We processed the daily sea ice concentrations for all studied glaciers from 1985 to 2023 and discussed the role of sea ice in both seasonal and long-term calving front changes.
- We investigated the large calving front change variability, as well as the high air and ocean temperature anomalies, in 2016 and 2019. We found they are likely driven by atmospheric blocking.
- We processed the CryoSat-2 CryoTEMPO-ELIOS gridded datasets from 2010 and 2023 but found the spatial coverage is very poor for marine-terminating glaciers we are interested in (see Figure R2 below). Therefore we decided to abandon the correlation analysis between annual glacier mass loss rates and calving front changes, and only focused on analyzing the overall ice discharge in Svalbard. We are planning on a separate paper on this relationship by improving Svalbard mass balance estimates from multi-source satellite data.
- We have made significant changes to the paper structure to accommodate these changes.

I know that Nature Comms is short format, which makes exploring ideas difficult, but unfortunately some of the arguments did not feel fully pinned down and explored, which makes the narrative less convincing. I think you have a really solid dataset here and some really interesting findings, so regardless of what is decided by Nature Comms, I strongly encourage submission elsewhere and this might give you the space to fix some of the issues noted above. Please see my annotated pdf for detailed comments.

Major comments

Line 29-31: I'm not sure about this statement: the Arctic has diverse oceanic / atmospheric / glaciological settings. I'd remove this and focus on what it tells us about glacier dynamics in an area with lots of ice that is highly responsive to climate change.

Thank you for this comment, agree and we have removed this statement.

Line 35-36: I would take surging out here, as it complicates things - it can obviously occur independent of climate but can also be linked. Given Svalbard has so many surge type glaciers, I'd make it clear you exclude them and focus only on the climate signal.

Agree and we have removed 'surging' from this sentence. In addition, we have made it clear throughout the manuscript that we excluded surging glaciers in the regional

analyses for seasonal and long-term calving front changes.

Line 37-28: And to some extent melt at the calving front - need to state somewhere how you separate this out and/or demonstrate that almost all of the ice loss is from calving (as opposed to enhanced melting across the ice front, e.g. due to plumes).

Good point, we made it clear throughout the manuscript that we measure calving front changes. In the same Section, we added a statement on why we use calving front changes instead of frontal ablation which requires ice velocity in Line 70-77 'We consider calving front advance as when ice discharge exceeds frontal ablation and retreat as when frontal ablation exceeds ice discharge. We are not able to quantify ice discharge, and therefore frontal ablation, due to a lack of ice velocity dataset of sufficient spatial coverage and temporal resolution for the entire Svalbard over a long period that is required for this study. However, calving front retreat rate has proved to be a good indicator for the variability of frontal ablation on Svalbard [12, 13]. In these studies where ice discharge has been taken into account, it is shown that frontal ablation in winter almost shut down, allowing most of the ice discharge to go into the seasonal readvance of the calving fronts.'

Line 40-42: So are you including this portion of ice loss? I.e. it's net ablation from all processes? This is more sensible, but needs to be made clear in the abstract, which talks about calving specifically, i.e. not net ablation at the front.

Yes, the calving front changes we observed include both calving and frontal melting. In the abstract, we now made it clear we measure calving front retreats in this study.

Line 46-47: There has been some recent work on this in NVZ. Please see (<https://www.cambridge.org/core/journals/journal-of-glaciology/article/rapid-and-synchronous-response-of-outlet-glaciers-to-ocean-warming-on-the-barents-sea-coast-novaya-zemlya/B1B061E7867A7FCDC863EF4EE2D946C9>) and Tepes references therein.

Thank you for the suggestion. It is a great piece of work on seasonal calving front changes on NVZ and we learned a lot from reading this paper. After checking the CryoSat-2 data, we decided to remove the relationship between dynamic mass loss and calving front area changes. There are two reasons for this change: 1) as reviewers 2 and 3 pointed out already, the current correlation is severely skewed by two data points in the plot which makes the correlation with Hugonnet data less convincing; 2) we checked the CryoSat-2 CryoTEMPO gridded datasets over Svalbard and found that the spatial coverage on marine-terminating glaciers is very poor (Figure R2); in addition, the time series of CryoSat-2 is only 12 years while we have 38 years of calving front observation. Therefore we think it is better to focus on discussing the integrated ice discharge over Svalbard from GRACE and CARRA datasets.

Line 48-49: Temporal or spatial? Please indicate the resolution here.

Good point, we have changed the sentence to 'Here we use a new calving front data

Figure R2: Spatial coverage of CryoSat-2 CryoTempo 2km gridded dataset, the red dots are the glacier points used in deriving MAR regional climate model variables for marine-terminating glaciers.

product over large spatial scales and at a high temporal resolution for 149 marine-terminating glaciers across Svalbard between 1985 and 2023' in Line 50-51.

Line 50-52: As noted above, to be calving is loss of ice through icebergs being removed from the front. It doesn't include melting e.g. due to plumes. You need to be consistent about which term you mean - total frontal ablation (i.e. calving + melt) is what I think you're measuring and is easier, as you don't have to separate out the two.

Agree, we made it clear now that we used calving front change which includes both calving and frontal melting throughout the manuscript.

Results Section: This seems to be a mixed results and discussion section. Either needs clearly separating or blending together fully. There are numerous points where the results move towards discussion.

Thank you for this suggestion, we have now merged the results with relevant discussions under different sections to better blend the whole story.

Line 57-59: So are you including surge type glaciers in here?? I think you should remove them, because this complicates the signal, particularly because of the long duration of surge cycles e.g. in Austfonna. Or, put another way, if you have the surging glaciers in here, how to do you determine what is climate driven versus internal glacier dynamics?

This is a very good point and we agree. In the revision, we have removed the surging glaciers from long-term analysis and only focused on non-surging marine-terminating glaciers. All the relevant figures, statistical analysis and text have been updated.

Line 63-64: Feels like this is drifting into discussion, unless they are meant to be combined?

Agree, we have now fully blended results with relevant discussion under individual section.

Line 68: This is nice novel work.

Thank you!

Line 71-73: Useful information. For interest, this is much less consistent than what we found on NVZ: there retreat usually began in June / July and advance in Nov / Dec and was really quite consistent between glaciers year.

Thank you for this comment. In the revision, we find a spatial variability in the timing of seasonal calving front retreat rates between west and east Svalbard, coincident with the spatial variability in subsurface ocean temperatures, likely driven by the delayed ocean warming along the east coast. This is likely why the seasonal cycles show a higher degree of inconsistency than NVZ. However, if we look at the time series of detrended calving front changes of individual glaciers from 2014 to 2023 in Figure 2, the timings of the retreat and advance between different glacier years are very consistent.

Line 73-75: Interesting

Thank you.

Line 81: remove would

Agree and done.

Line 84-87: I think you need to make it really clear here that these are REGULAR seasonal variations, i.e. they happen every year and are imprinted on top of the surge, rather than e.g. short-term surge events. If they are regular and mirror the timings you see on other glaciers, this is an interesting finding, hence why I think it really needs to be properly pinned down.

Thank you for this comment. Yes, these are regular seasonal cycles that happened every year and on top of the surging events. Please see Figures 2e and 2f for examples of two surging glaciers Vestre Ocbornebreen and Storisstraumen. We have also mentioned this in Line 87-93: "The seasonal calving front changes of surging glaciers follow similar patterns to non-surging glaciers (Figure S9b), although their average seasonal range is significantly higher, reaching 192 ± 54 m. Figures 2e and 2f show the seasonal calving front time series of two surging glaciers Vestre Osbornebreen (RGI60-07.00468) and Storisstraumen (RGI60-07.00027), respectively, where calving fronts usually retreat from July until December with a regular cycle. The similarity in seasonal cycles between surging and non-surging glaciers provides further evidence that dynamic regimes have minimal influence on the calving front cyclicity [12]."

Figure 1a: I'm being picky but this legend doesn't match the figure, i.e. the circles are red and blue, this has white circles. I'm assuming blue = advance and red = retreat, but should be in the legend.

In this figure, the size of the circle denotes the magnitude of the decadal front change rate, the blue circle denotes the advancing glacier and the red circle denotes the retreating glacier. We have added a label for colors in Figure 4a and included this information in the figure caption.

Figure 1c: This has really dropped off in recent years - perhaps it is discussed later, but interesting to know why.

Thanks for pointing this out. The drop in area loss rates in 2019 happened in different parts of Svalbard and in sync with drops in air and ocean temperatures (Figures 5 and 6b), likely driven by atmospheric blocking that slows down the large-scale ocean circulation in the Nordic Seas discovered by [5]. This has now been discussed in the Section 'Rapid glacierresponse to climate extremes'.

Figure 2: I like this figure overall, but a few suggestions:

- 1) The squares and colour range don't work for me - I find it really hard to see the pale squares. Suggest either circles or + and a different set of colours that are more obvious at the lower end of the scale.

- 2) Pale bars on the charts are hard to see. Suggest using a different colour, but a bolder one. Line 106-108: I'd add the spatial resolution here - e.g. do you have one grid square per glacier or 10 glaciers per grid square??

Thank you for the comments. We have now 1) changed squares into circles with a different set of colors; 2) changed the colors of histogram charts into bolder colors. This figure is now Figure 1.

Line 108-109: Same for this. To what extent do we actually have different data for different glaciers, given the resolution of these forcing datasets?

Thank you for this comment. For extracting variables from MAR regional climate model, the glacier point is generated as the center point of each glacier domain in [14] which means the locations of the glacier points for different glaciers are very different (Figure S5), this applies to the derived MAR variables. For ORAS5 data, the spatial resolution is 0.15° in Svalbard and the spatial coverage near the coast is low (Figures S6), this has resulted in some fjord points are located in the outer fjords, as pointed out by Reviewer 4. This, however, does not influence our conclusion on the relationship between ocean warming and glacier retreat rates at different timescales, because (also see our response to Reviewer 4) 'The fjord points in the open ocean are representative of the water structures within the fjord, as they are predominantly located in regions of deep bathymetry (Figure S6), indicating that these fjord points lie along the pathway where warm Atlantic water can enter the fjords. Furthermore, in-situ measurements in west Spitsbergen show that ocean temperatures in water depths less than 100 m are similar both inside and outside the fjords [15, 16].' in Line 106-111.

Line 117-119: What about sea ice? Do you think this may be important here? Or, put another way, do the switches from advance to retreat (and visa versa) coincide with sea ice formation / disintegration?

Good point. We have now analyzed seasonal changes in sea ice concentration for 86 non-surging marine-terminating glaciers in Svalbard by using daily sea ice concentrations from AMSR-E/2 available from the University of Bremen and their relationship with seasonal calving front changes over the period of 2014-2023. We also plotted the monthly sea ice concentration changes for specific glaciers in Figure 2 and calculated the regional monthly sea ice concentration and sea-ice-free years shown in Figure S8.

We find "Sea ice concentration ($R^2=0.83$) is the second strongest predictor for seasonal front retreat rates. Landfast sea ice (hereafter referred to as fast ice) can exert a buttressing effect on glacier calving fronts, stabilizing the ice and reducing calving rates [17]. However, in Svalbard, there is a large spatial variability in seasonal sea ice concentration (Figure S8), and glaciers in Northwest and South Spitsbergen are largely sea ice free in winter over the 2014-2023 period (Figures 2a,2b and 2e). Eastern Svalbard including Austfonna, Vestfonna, Barentsøya and Edgeøya, feature higher sea ice concentration. The average sea ice concentrations in early Spring in Austfonna and Vestfonna are around 70%, while it can exceed 80% in Barentsøya and

Edgeøya (Figure S8), which has the highest correlation with sea ice (Table S1). This spatial pattern matches the fast ice distribution in the past two decades which primarily concentrated along the coast of eastern Svalbard [18]." now in Line 131-140.

Line 119-120: First, this is really discussion. There are several points where this happens - results should just state what you found. Second, how appropriate is this reference to use - it's for Jakobshavn Isbrae, which used to have a large floating tongue, so is perhaps not comparable to the glaciers you have here.

Thank you for the comment and agree. We have now removed this sentence.

Line 123-125: How does this compare to when ocean temperatures increase? Are we sure this is just air temperatures that are changing??

Good point, the onset of seasonal calving front retreats are coincident with both surface runoff and subsurface ocean temperature. We have rephrased this sentence as "The onset of calving front retreat coincides with the increased surface runoff and rising subsurface ocean temperature, while the peak retreat rates occur during the months of highest ocean temperatures (Figure 1d)." in Line 82-84.

Line 130-131: What do you define as 'abnormal'? A certain standard deviation above the mean?

Thank you for the comment, this is a very good point. First, we have removed word 'abnormal'. Second, a recent study on climate extremes in Svalbard over the last two millennia flagged the high air temperature in 2016 as 'climate extreme' [6]. As for 2019, we can see that both air and subsurface ocean temperatures significantly dropped around 6 °C and 3 °C compared to 2016 (Figure 6b). This significant temperature cooling has been discussed in a recent study [5] to be related to the slowdown of large-scale ocean circulation in the Nordic Seas driven by atmospheric blocking.

Line 130-132: As above, what does the normal state mean? The long term average. Try to be precise in the definition of these terms.

Agree, we have changed the wording from 'the normal state' to 'pre-2016 level' in Line 210.

Line 132: 'Svalbard-wide mean cumulative area losses', Including the surge glaciers??

It did not include surging glaciers in the initial submission. In the revision, we have made it clear that for long-term glacier retreats we only consider non-surging marine-terminating glaciers.

Line 135-137: But you haven't analyzed sea ice data, so how do you know it's important? Also, this is again moving towards discussion.

Thank you for the comment, we have now added sea ice concentration analysis and changed this statement to "Similar to seasonal front changes, the highest correlation is found for subsurface ocean temperature ($R^2=0.5$), followed by air temperature ($R^2=0.39$) and sea ice concentration ($R^2=0.34$), suggesting that ocean forcing is the most prominent factor in controlling the interannual calving front variability. However, air temperature also shows a comparable pattern and correlation, as well as the decreased sea ice cover and increased occurrences of sea-ice-free months around Svalbard (Figure S14)." in Line 185-189.

Line 140-142: But you said above there was also a correlation with air temperatures with a month lag, so how does that fit with this statement? What about ocean temperatures? Can we rule out their impact, e.g. are they low at this time of the year?

Thank you for this comment. We have now removed the analysis between one-month lagged air temperature and seasonal calving front changes as this information no longer fits into our storyline in the revision.

Line 164-165: Useful figure

Thank you.

Line 165-167: Drifting into discussion.

We have changed the paper structure to fully blend results and associated discussion in each section.

Line 167: Give the % split as this is an important point. Please comment on the year to year variability, e.g. is this increase mainly due to the big spike in ice loss around 2018? Has the proportion reduced in later years? I think it is important to note this variability in contribution from glacier dynamics. ALSO, are surges removed from this, as they will complicate the picture, e.g. if you have a large surge event in one year and net advance of a large basin (e.g. Basin 3) then you could really effect the numbers for that year.

Thank you for this comment. We have decided not to present dynamic imbalance as a key conclusion in the paper anymore. However, we believe that the increased discharge over the past two decades remains crucial for providing context on the broader dynamic mass loss associated with retreats of marine-terminating glaciers in Svalbard. We are not able to isolate the mass loss related to surge events from the ice discharge calculated from GRACE mascons and CARRA SMB data, as 1) it is difficult to identify all the surging events happening in Svalbard during our study periods; 2) therefore measuring the mass loss of surging events requires a significant amount of extra work which is more suitable for a separate paper on this topic. Given these two reasons, we decided not to investigate the interannual variability in the glacier percentage changes in this study.

Line 168-170: I'd also use one of the many glacier inventories noting surging as you might not pick them all up from this, given Svalbard's long surge times.

Thank you for this comment, this is a very good advice. However, we think the current analysis of Hugonnet mass loss rates by using the same non-surge type glaciers from our study is more appropriate compared to using a global surging glacier inventory such as RGI. The reason is surging events mostly happen in a very short time period that is difficult to capture, so there will be a mismatch between the surge glaciers identified by RGI and the surge events flagged by our study based on satellite image analysis.

Line 173-175: Doesn't necessarily have to be via common climate drivers - there could be a retreat due to ocean temperatures in one area and air temps in another, but still correlates. Also, please briefly outline the evidence for the direction of this relationship. i.e. what is the evidence that glacier retreat is causing negative surface mass balance or ice loss (presumably via dynamic thinning) versus negative SMB causing retreat (e.g. by reducing ice delivery and thinning near terminus) regions)

Thank you for this comment. We have removed 'via common climate drivers'. As mentioned in our responses to previous comments that we have analyzed the 12 years of CryoSat-2 CryoTempo time series across Svalbard but we found that the spatial coverage is very poor for marine-terminating glaciers we are interested in (Figure R2). Therefore we think using CryoSat-2 data will introduce very high uncertainties. We think the way to move forward is to improve mass changes in Svalbard by combining different satellite measurements including altimetry and gravimetry, a topic to be explored in our future study.

Figure 5: I think this correlation is getting really skewed by the top two points, If you take this out, I don't think you'll see much of a correlation. Unfortunately this is common with glacial datasets (i.e. the big glaciers dominate) but I don't think you can say from this plot that for all glaciers there is a strong correlation between retreat and total mass loss, as those top points are strongly impacting the relationship.

Thank you for this comment. We agree that this correlation is skewed by the top two points and we decided to remove this correlation analysis from the manuscript.

Line 186-188: I'm not sure it does tell you that this process is important. It might be, but other processes may also contribute, e.g. undercutting at the water line. Do you have additional evidence, e.g. of meltwater plumes that would support this idea?

Agree, we have added new analyses to better prove the role of ocean warming in controlling seasonal calving front changes. We have modified this sentence into "This regional variation in peak retreat rates closely aligns with the seasonal subsurface ocean temperature patterns across different regions, suggesting that ocean temperature exerts an immediate impact on calving front retreat, as oceanic forcing can manifest the submarine melting of glaciers and increase glacier retreat rates by combined

undercutting and calving [19, 20, 4, 21, 13]." in Line 116-120.

Line 190-192: How would this accumulated effect work? Are we saying that we need fractures to fill up enough or is it e.g. repeated opening and reclosing until a critical stress is reached?

Thank you for the comment, we have removed this sentence from the manuscript.

Line 192-193: As above, is there evidence to support this here, e.g. plumes at the front?

Thank you for the comment, we have removed this sentence from the manuscript.

Line 195-197: How are they preconditioned? This also brings me back to the point above - is the surface lowering we're seeing a cause of retreat or driven by it? This isn't an easy question. You may find some useful thoughts on this in the following paper, which tries to untangle this relationship for NVZ. <https://www.cambridge.org/core/journals/journal-of-glaciology/article/rapid-and-synchronous-response-of-outlet-glaciers-to-ocean-warming-on-the-barents-sea-coast-novaya-zemlya/B1B061E7867A>

Thank you for this comment, and we agree this is not an easy question to solve which requires improving the current elevation change and mass balance measurements in Svalbard. We have decided to remove this correlation analysis from the revision.

Line 201-202: This feels out of place here. The point about seasonal cycles still matching even between surge and non-surge type glaciers is useful finding, but feels lost here.

Agree, we have now added new analyses on the seasonal calving front changes for eight surging glaciers that exhibit seasonal cycles in our study, this information is now included in Section 'Widespread seasonal calving front changes across Svalbard'.

Line 206-207: In the discussion, I think you should note the massive spike in the late 2010s and explain it. You should also discuss the subsequent reduction in retreat, which is also really worth noting and goes against general trends.

Thank you for this valuable comment. We have analyzed the calving front area changes in years 2016 and 2019 and their links to significant temperature changes in these two years, which is likely driven by atmospheric blocking. Relevant information are now in Section 'Rapid glacier response to climate extremes'.

Line 211-213: Reference the figure or paper showing this.

Thank you for this comment, we have deleted this sentence from the paper.

Line 216: Be consistent in reference style - should be numbered.

Agree and done.

Line 224-226: I'm not sure I agree with this. As noted above, I think you are getting this correlation because of the two largest retreats. i.e. this isn't a pervasive relationship across all of the glaciers.

Agree, we have removed this part from the manuscript as mentioned in our previous responses.

Reviewer 4

Review of Atmospheric and oceanic warming drives pervasive retreat of marine-terminating glaciers in Svalbard by Li et al.

General Comments

This is an interesting article which relates environmental variables to glacier retreat on a pan-Svalbard scale. Overall, I think the manuscript is well written and scientifically sound. However, I have one general comment related to the location of the fjord points used for extracting the oceanographic data. In the supplementary material, Fig S5 shows the fjord points used. Many of these fjord points are a considerable distance from the glacier fronts, and even outside of the fjord mouth. Previous research has shown that the water masses present in the inner fjords can be quite different to those present outside of the fjord mouth (see e.g. Fig. 4 in Prominska et al, <https://doi.org/10.1016/j.oceano.2017.07.003> and/or Divya et al <https://doi.org/10.1016/j.polar.2021.100735>). I understand that the location of these fjord points may be an unavoidable consequence of the pan-Svalbard scale of the study / the datasets used, but feel that this needs to be addressed in the main text. For example, a short paragraph could be added to the discussion.

We would like to thank the reviewer for the positive and constructive comments, which have greatly improved this paper.

As the reviewer has noticed that some of the fjord points generated in our study are located a bit far away from the glaciers. The reason for this spatial deviation is caused by the limited spatial coverage of the ORAS5 data in the coast areas. We generated the fjord points for marine-terminating glaciers around Svalbard by finding the closest non-empty grid point in the ORAS5 grid to the corresponding glacier point of each glacier derived from MAR data (see Online Methods). Despite this, we argue that the fjord points in the outer fjords are still representative of the ocean water structures inside the fjord. In Figure S6, we can see that the majority of fjord points are located in regions where the bathymetry is deep, indicating that they are in the pathway where the warm Atlantic water can enter the fjords. In addition, the CTD samplings in several fjords in west Spitsbergen in [15, 16] show that the ocean temperatures in water depths less than 100 m outside and inside the fjords are quite similar. A short discussion has been added to the Section 'Ocean temperature variability drives divergent seasonal cycles' in Line 102-111:

"The spatial distributions of the fjord points used to extract ocean temperatures from ORAS5 ocean reanalysis data are shown in Figure S6. Although some fjord points are far from glacier terminus and sample ocean temperatures outside fjords due to limited spatial resolution and coverage of ocean reanalysis data near the coast, this does not impact the relationship between ocean warming and seasonal calving front retreats. The fjord points in the open ocean can still be representative of the water structures within the fjord, as they are predominantly located in regions of deep bathymetry (Figure S6), indicating that these fjord points lie along the pathway where warm Atlantic water can enter the fjords. Furthermore, in-situ measurements in west Spitsbergen show that ocean temperatures in water depths less than 100 m are often similar both inside and outside the fjords [15, 16]."

In addition, I have several minor comments which are listed below:

Minor comments

L38: I would appreciate a little more context here: is frontal ablation not considered in other studies, or assumed to be included in some other metric etc?

Thank you for this comment. We have rephrased this sentence as "Glaciers in Svalbard have been losing mass at accelerated rates during the past several decades [22, 23, 24, 25], their mass loss is expected to double by 2100 but with significant uncertainties in the predicted rates [1], partly due to a lack of consideration of retreats of marine-terminating glaciers [2, 3] which drain ~69% of the glacierized area [26]." in Line 34-37.

L60: 'Area at the calving front' – I see this explained in the methods, but was not fully sure what this meant until I found this explanation. Maybe a short explanation can be included in the text here

Thank you for this comment. We have added a sentence to explain the area changes at glacier calving fronts: "The area change is calculated by multiplying the calving front length change by the glacier width, defined as the mean length of all terminus traces (see Online Methods)" in Line 153-154.

L64: I appreciate the comparison with Kochitsky et al, as well as the possible explanation for the difference between their results and yours. I wonder if you would get a more similar result if you used lower (temporal) resolution data / have you tried this and was this the outcome? Then it would be clear if this was the reason for the difference in results, or suggest that something else is at play

Thank you for this comment, this is a very good point. We have now compared the area changes between our study and the Kochitzky 2022 dataset using a common set of glaciers over two periods: 2000-2010 and 2010-2020. The results are shown in Figures S11 and S12 and in Table S3. We find that the mismatch in area changes mainly comes from five large glaciers with large glacier widths in the period of 2000-

2010 (Figure S12), so a slight mismatch in the calving front measurements will result in large differences in area changes. After removing these five large glaciers, the difference in area changes between these two datasets is only 49 km^2 between 2000 and 2020. We have now added a discussion on area change comparison in Line 154-164.

L78: It is not clear to me whether your description of the R^2 values as significant means you think this shows a strong relationship, or whether you just mean the result is statistically significant. Related to this, it would be useful if you state what you consider a 'significant' R^2 value to be, and also what the p values are for each R^2 value. A value of 0.37 is perhaps quite good for a natural system, but is much lower than any of your R^2 values linking retreat rates to environmental values, so some discussion of how you interpret the R^2 values would be beneficial.

Thank you for this comment. In our initial submission, we mean 'significant' as statistically significant because of a low P value (<0.05). To avoid confusion, we have rephrased this sentence as "The range of seasonal calving front variability correlates with the decadal retreat rates with a correlation coefficient R value of 0.6 ($P<0.05$, Figure S13a), suggesting that glaciers experience a larger amplitude in the seasonal cycle also experience a greater amplitude of secular trend." in Line 165-167.

L83: Do you have any suggestions for why the seasonal range/long-term range R^2 value is twice as high for Greenland?

Thank you for the comment. Please see our response to the comment made by Reviewer 1:

We would like to admit that our previous conclusion regarding the correlation between these two variables in Svalbard being lower than in Greenland Ice Sheet was incorrect. In fact, the correlations are very similar. The R^2 value of 0.71 for the Greenland Ice Sheet in [7] was calculated between the long-term mass loss and seasonal mass loss, not the effective length changes that we measured in Svalbard. In the **Extended Data Fig. 4** in [7], they normalized the terminus mass change values by terminus area and calculated the correlation between seasonal range and long-term changes in length variability, the **R-value** is only 0.47, which is slightly lower than the R-value of 0.6 ($P<0.05$) for Svalbard (Figure S13a). Therefore, we confirm that the relationship is nearly identical between Svalbard and Greenland. We have corrected this statement in Line 169 as 'This finding agrees with a recent study of glacier retreats in the Greenland Ice Sheet'.

L113: Interesting that the r^2 value for subsurface ocean temperatures is very similar to that for one month lagged air temperatures – do the time series for these two variables look similar?

Thank you for the comment. The time series of mean air temperatures and ocean temperatures are similar on the Svalbard scale. However, we have now removed this

statement from the manuscript, our new analysis on regional seasonal calving front retreats shows that the spatial variability of seasonal calving front retreats coincident with changes in ocean warming while the timings of peak air temperatures are similar across different regions. This directly demonstrates the dominant role of ocean warming in driving seasonal calving front changes.

L120: Please state whether this is basal melt under grounded or floating ice, and also specify if this is an additional 15 m/yr of basal melting or an absolute value.

Thank you for the comment, we have now removed this sentence according to the comments made by Reviewers 1 and 3.

L124: You could link this to e.g. Slater and Straneo (<https://www.nature.com/articles/s41561-022-01035-9>), where atmospheric temps was found to exert a first order control on submarine melt

Thank you for this suggestion, we have added a statement on the role of atmospheric warming on seasonal calving front changes by linking the Slater and Straneo (2022) along with other relevant studies in Lines 136-145:

"Evidence suggests that both these factors can influence calving rates through enhanced hydrofracture at the surface and subaqueous thermohaline circulation beneath the floating ice [4]. Surface meltwater caused by atmospheric warming can increase subglacial discharge which can lead to increased basal melting, as it can create buoyant plumes that enhance the turbulent transfer of oceanic heat to glacier calving fronts [27, 28, 29, 21], while the accumulation of surface meltwater contributes to crevasse hydrofracturing [30, 31, 32]. The summer melt season in Svalbard usually lasts about six months between May and October (Figure 1d) and the onset of surface runoff in May coincides with the most frequent starting month for frontal retreat (Figures 1b-d). Similar relationships are also found for surge-type glaciers (Figure S9)."

Fig. 3: Please add p values to the annotation on the plots (the same for other figures)

Agree and done for all the figures.

L175: Could it also be some internal dynamic feedbacks? E.g. climate drivers cause front retreat, which cause acceleration and then thinning etc?

Yes it can be internal dynamic feedback between calving front retreat and glacier acceleration and thinning. We have removed this sentence from the manuscript, please also see our responses to Reviewer 3.

L204: I think this should be 'mirroring a recent finding FROM/FOR THE Greenland Ice Sheet'

Thank you for the suggestion, but we have rephrased this sentence in Line 169-171:

"This finding agrees with a recent study of glacier retreats in the Greenland Ice Sheet [7], which identified seasonal length variability as the sole significant predictor of long-term calving front change ($R = 0.47$, Extended Data Fig. 4 in [7]) among various candidate predictors".

L205: Reference should be in brackets

Agree and revised.

L318: Can you add a comment on the impact of the resolution of the ocean data set, given the known importance of fjord circulation for submarine melt rates

Thank you for this comment. We rechecked the spatial resolution of ORAS5 data and found the horizontal resolution is approximately 0.15° in irregular grids in Svalbard, we have updated this number instead of directly citing the value provided in [33]. We have regridded the data to regular grids of 0.15° , relevant information in Online Methods have been updated. We added a statement in the Online Methods: 'This spatial resolution is enough to capture the spatiotemporal variability of ocean temperature outside the fjords as the width of the fjord mouth is normally larger than the grid size of ORAS5 data' in Line 349-352.

References

- [1] E. C. Geyman et al. Historical glacier change on svalbard predicts doubling of mass loss by 2100. *Nature*, 601:374–379, 2022.
- [2] T. V. Schuler et al. Reconciling svalbard glacier mass balance. *Frontiers in Earth Science*, 8:156, 2020.
- [3] D. R. Rounce et al. Global glacier change in the 21st century: Every increase in temperature matters. *Science*, 379:78–83, 2023.
- [4] D. A. Slater and F. Straneo. Submarine melting of glaciers in greenland amplified by atmospheric warming. *Nature Geoscience*, 15(10):794–799, October 2022. ISSN 1752-0908. doi: 10.1038/s41561-022-01035-9.
- [5] Rebecca Adam McPherson, Claudia Wekerle, Torsten Kanzow, Monica Ionita, Finn Ole Heukamp, Ole Zeising, and Angelika Humbert. Atmospheric blocking slows ocean-driven melting of greenland's largest glacier tongue. *Science*, 385(6715):1360–1366, September 2024.
- [6] Francois Lapointe, Ambarish V Karmalkar, Raymond S Bradley, Michael J Retelle, and Feng Wang. Climate extremes in svalbard over the last two millennia are linked to atmospheric blocking. *Nat. Commun.*, 15(1):4432, June 2024.
- [7] C. A. Greene, A. S. Gardner, M. Wood, and J. K. Cuzzone. Ubiquitous acceler-

- ation in greenland ice sheet calving from 1985 to 2022. *Nature*, 625:523–528, 2024.
- [8] Denis Felikson, Ginny A. Catania, Timothy C. Bartholomaus, Mathieu Morlighem, and Brice P. Y. Noël. Steep glacier bed knickpoints mitigate inland thinning in greenland. *Geophysical Research Letters*, 48:e2020GL090112, 2021. doi: 10.1029/2020GL090112.
- [9] Stewart Jamieson, Andreas Vieli, Stephen Livingstone, et al. Ice-stream stability on a reverse bed slope. *Nature Geoscience*, 5(11):799–802, 2012. doi: 10.1038/ngeo1600.
- [10] G. Hilmar Gudmundsson, J. Krug, G. Durand, L. Favier, and O. Gagliardini. The stability of grounding lines on retrograde slopes. *The Cryosphere*, 6(6):1497–1505, 2012. doi: 10.5194/tc-6-1497-2012.
- [11] Ginny A. Catania, Leigh A. Stearns, David A. Sutherland, Mason J. Fried, Timothy C. Bartholomaus, Mathieu Morlighem, et al. Geometric controls on tidewater glacier retreat in central western greenland. *Journal of Geophysical Research: Earth Surface*, 123:2024–2038, 2018. doi: 10.1029/2017JF004499.
- [12] A. Luckman et al. Calving rates at tidewater glaciers vary strongly with ocean temperature. *Nature Communications*, 6:8566, 2015.
- [13] Ø. Foss, J. Maton, G. Moholdt, L.S. Schmidt, D.A. Sutherland, I. Fer, F. Nilsen, J. Kohler, and A. Sundfjord. Ocean warming drives immediate mass loss from calving glaciers in the high arctic. *Accepted in Nature Communications*, 2024.
- [14] T. Li et al. A high-resolution calving front data product for marine-terminating glaciers in svalbard. *Earth System Science Data*, 16:919–939, 2024.
- [15] Agnieszka Promińska, Małgorzata Cisek, and Waldemar Walczowski. Kongsfjorden and hornsund hydrography – comparative study based on a multiyear survey in fjords of west spitsbergen. *Oceanologia*, 59(4):397–412, 2017. ISSN 0078-3234. doi: 10.1016/j.oceano.2017.07.003.
- [16] David T. Divya, M.P. Subeesh, V.S. Anju, and N. Anilkumar. Variability in the hydrography of two proximate arctic fjords during 2013–18. *Polar Science*, 30:100735, 2021. ISSN 1873-9652. doi: 10.1016/j.polar.2021.100735.
- [17] T. Moon, I. Joughin, and B. Smith. Seasonal to multiyear variability of glacier surface velocity, terminus position, and sea ice/ice mélange in northwest greenland. *Journal of Geophysical Research: Earth Surface*, 120:818–833, 2015.
- [18] Jacek A. Urbański and Dagmara Litwicka. The decline of svalbard land-fast sea ice extent as a result of climate change. *Oceanologia*, 64(3):535–545, 2022. ISSN

0078-3234. doi: <https://doi.org/10.1016/j.oceano.2022.03.008>.

- [19] D. M. Holland, R. H. Thomas, B. De Young, M. H. Ribergaard, and B. Lyberth. Acceleration of jakobshavn isbr triggered by warm subsurface ocean waters. *Nature Geoscience*, 1:659–664, 2008.
- [20] F. Straneo et al. Challenges to understanding the dynamic response of greenland’s marine terminating glaciers to oceanic and atmospheric forcing. *Bull. Am. Meteorol. Soc.*, 94:1131–1144, 2013.
- [21] Claudia Wekerle, Rebecca McPherson, Wilken-Jon von Appen, Qiang Wang, Ralph Timmermann, Patrick Scholz, Sergey Danilov, Qi Shu, and Torsten Kanzow. Atlantic water warming increases melt below northeast greenland’s last floating ice tongue. *Nature Communications*, 15(1), February 2024. ISSN 2041-1723. doi: 10.1038/s41467-024-45650-z.
- [22] R. Hugonnet et al. Accelerated global glacier mass loss in the early twenty-first century. *Nature*, 592:726–731, 2021.
- [23] P. Tepez et al. Changes in elevation and mass of arctic glaciers and ice caps, 2010–2017. *Remote Sensing of Environment*, 261:112481, 2021.
- [24] C. Nuth et al. Svalbard glacier elevation changes and contribution to sea level rise. *Journal of Geophysical Research*, 115:F01008, 2010.
- [25] A. Morris et al. Spread of svalbard glacier mass loss to barents sea margins revealed by cryosat-2. *Journal of Geophysical Research: Earth Surface*, 125: e2019JF005357, 2020.
- [26] J. J. Fürst et al. The ice-free topography of svalbard. *Geophysical Research Letters*, 45:11760–11769, 2018.
- [27] T. R. Cowton, A. J. Sole, P. W. Nienow, D. A. Slater, and P. Christoffersen. Linear response of east greenland’s tidewater glaciers to ocean/atmosphere warming. *Proc. Natl. Acad. Sci. U. S. A.*, 115:7907–7912, 2018.
- [28] S.J. Cook, P. Christoffersen, M. Truffer, T.R. Chudley, and A. Abellán. Calving of a large greenlandic tidewater glacier has complex links to meltwater plumes and mélange. *Journal of Geophysical Research: Earth Surface*, 126(4):e2020JF006051, 2021.
- [29] E. Rignot, M. Koppes, and I. Velicogna. Rapid submarine melting of the calving faces of west greenland glaciers. *Nature Geoscience*, 3(3):187–191, 2010.
- [30] D. I. Benn, C. R. Warren, and R. H. Mottram. Calving processes and the dynamics of calving glaciers. *Earth-Science Reviews*, 82:143–179, 2007.

- [31] J. Downs, D. Brinkerhoff, and M. Morlighem. Inferring time-dependent calving dynamics at helheim glacier. *J. Glaciol.*, 69:381–396, 2023.
- [32] C. J. Van Der Veen. Tidewater calving. *J. Glaciol.*, 42:375–385, 1996.
- [33] H. Zuo, M. A. Balmaseda, S. Tietsche, K. Mogensen, and M. Mayer. The ecmwf operational ensemble reanalysis-analysis system for ocean and sea ice: A description of the system and assessment. *Ocean Science*, 15:779–808, 2019.

Reviewer #3 (Remarks to the Author):

I feel that the authors have provided a comprehensive, effective and well-considered response to my comments. I am satisfied by their responses and think they have a really nice paper with an excellent dataset, that provides novel insights into Arctic glacier behaviour. They should be commended on a really nice piece of work and their throughout response to reviewer comments. I have no further comments and look forward to seeing the final version.

Thank you for taking the time to review our paper and for your positive comments on our revision.

Reviewer #4 (Remarks to the Author):

Thank you to the authors for your comprehensive responses to all the reviews, and the associated changes in the manuscript. The edits have, in my opinion, led to significant improvements and I do not have any major concerns with the article in its current form.

Thank you for taking the time to review our paper and for your positive comments on our revision.

However, I would ideally like to see your statement regarding the ocean temperature dataset changed. I appreciate you adding in more information about the dataset but am not fully convinced by the statement that the distance of data points from the coast 'does not impact the relationship between ocean warming and seasonal calving front retreats'. Data/figures from e.g. Promińska et al that show AW in your chosen depth range is present in the outer fjord but not near the glacier front in several years (e.g. 2002, 2007, 2013, 2014). Other previous research has showed that the relationships between ocean temps and frontal ablation is different when you take ocean data from closer to the calving front (Holmes et al.). Whilst I am onboard with your approach being appropriate given the scale of the study, the statement quoted above needs to be toned down.

Thank you for this suggestion and we agree. We have changed this statement to 'this does not significantly affect the relationship between ocean warming and seasonal calving front retreats'.

One minor comment is that your units are sometimes italicized (e.g. L164) but sometimes not (e.g. L169) – this should be consistent throughout the manuscript.

Thank you for the suggestions, we have changed the style of the unit to italicized throughout the manuscript.